EMBO
Molecular Medicine

# Modeling, optimization, and comparable efficacy of T cell and hematopoietic stem cell gene editing for treating hyper-IgM syndrome

Valentina Vavassori[1,2,†], Elisabetta Mercuri[1,3,†], Genni E Marcovecchio[1], Maria C Castiello[1,4], Giulia Schiroli[1], Luisa Albano[1], Carrie Margulies[5], Frank Buquicchio[5], Elena Fontana[4,6], Stefano Beretta[1], Ivan Merelli[1,7], Andrea Cappelleri[8,9], Paola MV Rancoita[10], Vassilios Lougaris[11], Alessandro Plebani[11], Maria Kanariou[12], Arjan Lankester[13], Francesca Ferrua[1,14], Eugenio Scanziani[8,9], Cecilia Cotta-Ramusino[5], Anna Villa[1,4], Luigi Naldini[1,2,*,‡] [iD] & Pietro Genovese[1,**,‡] [iD]

## Abstract

Precise correction of the *CD40LG* gene in T cells and hematopoietic stem/progenitor cells (HSPC) holds promise for treating X-linked hyper-IgM Syndrome (HIGM1), but its actual therapeutic potential remains elusive. Here, we developed a one-size-fits-all editing strategy for effective T-cell correction, selection, and depletion and investigated the therapeutic potential of T-cell and HSPC therapies in the HIGM1 mouse model. Edited patients' derived CD4 T cells restored physiologically regulated CD40L expression and contact-dependent B-cell helper function. Adoptive transfer of wild-type T cells into conditioned HIGM1 mice rescued antigen-specific IgG responses and protected mice from a disease-relevant pathogen. We then obtained ~ 25% *CD40LG* editing in long-term repopulating human HSPC. Transplanting such proportion of wild-type HSPC in HIGM1 mice rescued immune functions similarly to T-cell therapy. Overall, our findings suggest that autologous edited T cells can provide immediate and substantial benefits to HIGM1 patients and position T-cell ahead of HSPC gene therapy because of easier translation, lower safety concerns and potentially comparable clinical benefits.

**Keywords** CRISPR-Cas gene editing; hematopoietic stem cells; T-cell therapy; truncated EGFR; X-linked hyper-IgM Syndrome
**Subject Categories** Genetics, Gene Therapy & Genetic Disease; Immunology

## Introduction

X-linked hyper-immunoglobulin M (IgM) syndrome (X-HIGM or HIGM1, OMIM#308230) is a primary immunodeficiency caused by inactivating mutations of *CD40LG* gene. CD40 ligand (CD40L) is a type II transmembrane glycoprotein member of the tumor necrosis factor (TNF) superfamily (Van Kooten & Banchereau, 2000), which is mainly expressed in a tightly regulated manner on the surface of activated CD4 T cells (Armitage *et al*, 1992; Roy *et al*, 1993), although some expression has also been reported on other hematopoietic cell types, including B cells, NK, CD8 T cells, and basophils (Van Kooten & Banchereau, 2000). The function of CD40L

1 San Raffaele Telethon Institute for Gene Therapy, IRCCS San Raffaele Scientific Institute, Milan, Italy
2 Vita-Salute San Raffaele University, Milan, Italy
3 Milano-Bicocca University, Monza, Italy
4 Institute of Genetic and Biomedical Research Milan Unit, National Research Council (CNR), Milan, Italy
5 Editas Medicine, Cambridge, MA, USA
6 Human Genome Lab, Humanitas Clinical and Research Center, Milan, Italy
7 Institute for Biomedical Technologies, National Research Council (CNR), Segrate, Italy
8 Mouse and Animal Pathology Laboratory (MAPLab), Fondazione Unimi, Milano, Italy
9 Department of Veterinary Medicine, University of Milan, Milan, Italy
10 University Center for Statistics in the Biomedical Sciences (CUSSB), Vita-Salute San Raffaele University, Milan, Italy
11 University of Brescia and ASST-Spedali Civili di Brescia, Brescia, Italy
12 First Department of Paediatrics, Aghia Sophia Children's Hospital, Athens, Greece
13 Department of Pediatrics, Leiden University Medical Center, Leiden, The Netherlands
14 Pediatric Immunohematology and Bone Marrow Transplantation Unit, IRCCS San Raffaele Scientific Institute, Milan, Italy
*Corresponding author. Tel: +39 02 2643 4681; E-mail: naldini.luigi@hsr.it
**Corresponding author. Tel: +1 617 582 9395; E-mail: pietro.genovese@childrens.harvard.edu
†These authors contributed equally to this work
‡These authors contributed equally to this work as senior authors
§Present address: Department of Pediatric Oncology, Gene Therapy Program, Dana-Farber/Boston Children's Cancer and Blood Disorders Center, Harvard Medical School, Boston, MA, USA

is best known for CD4 T cells, which upon activation engage its cognate CD40 receptor on antigen presenting cells (APCs), such as B cells and macrophages, and stimulate their activation and maturation. In antigen-activated B cells, CD40L stimulation triggers proliferation, germinal center formation, antibody affinity maturation, and class-switching and long-term memory responses (Van Kooten & Banchereau, 2000). In monocytes, macrophages, and dendritic cells, CD40L stimulation enhances survival and killing and activates production of cytokines such as IL-1, IL-12, and TNF-α (Van Kooten & Banchereau, 2000). Less is known about CD40L function in other cell types, including a subset of CD8 memory T cells, which functionally resemble CD4 helper T cells, basophils, whose CD40L contributes to IgE switching in engaged B cells, and platelets, which release a soluble CD40L isoform upon activation (Henn *et al*, 1998; Yanagihara *et al*, 1998; Frentsch *et al*, 2013).

HIGM1 patients, although presenting normal number of T cells (Jain *et al*, 1999), invariably show hypogammaglobulinemia, with normal or elevated levels of serum IgM, due to impaired production of switched Ig isotypes. They characteristically present a high susceptibility to bacterial, intracellular, and opportunistic pathogens (mainly *Pneumocystis jiroveci* and *Cryptosporidium* spp.) and may develop biliary tract and liver disease, neutropenia, autoimmunity, and malignancies (Qamar & Fuleihan, 2014). Despite conservative therapies based on immunoglobulins supplementation and antibiotic prophylaxis, long-term survival is poor, with an average time from diagnosis of 25 years (de la Morena *et al*, 2017; Ferrua *et al*, 2019). Allogeneic hematopoietic stem cell transplant (HSCT) is the only curative treatment currently available, with best outcome obtained when performed early after diagnosis, before 10 years of age, in the absence of pre-existing organ damage (mainly liver disease) (Bucciol *et al*, 2019; Ferrua *et al*, 2019). Use of myeloablative regimens and matched donors (both siblings and unrelated) are recommended to achieve superior overall survival (Ferrua *et al*, 2019). However, matched donors are not always available, and HSCT still remains associated with risk of graft rejection, graft vs. host disease (GvHD), infections, liver failure, and death (Ferrua *et al*, 2019). Thus, therapeutic alternatives to safely and more effectively treat patients for whom HSCT is too risky are strongly needed.

Since HIGM1 is a monogenic disorder, gene therapy with autologous hematopoietic stem/progenitor cells (HSPC) corrected by gene replacement vectors was explored in pre-clinical studies. Despite these studies showed that a relatively low proportion of transduced HSPC could partially restore humoral and cellular immune function in a HIGM1 mouse model, constitutive and unregulated expression of CD40L in their thymocytes and peripheral T cells progeny led to lymphoproliferative disorders that were independent from vector integration and most of which progressed to lymphomas (Brown *et al*, 1998). Similarly, when the transgene was driven by a T-cell restricted promoter, its unregulated expression severely perturbed the homeostasis of the lymphoid subsets in lymph nodes and generated a hyperplastic B-cell expansion with high risk of progression to lymphoma (Sacco *et al*, 2000). Gene correction strategies that reconstitute physiological expression regulation of the corrected gene therefore represent more suitable approaches for the treatment of HIGM1 (Tahara *et al*, 2004; Romero *et al*, 2011).

Over the last years, targeted gene editing has emerged as potential therapeutic option for several genetic diseases (Genovese *et al*, 2014; Dever *et al*, 2016; De Ravin *et al*, 2017; Schiroli *et al*, 2017).

By exploiting engineered site-specific nucleases, such as zinc-finger nucleases (ZFN), TAL effector nucleases or CRISPR/Cas9, it is possible to deliver a DNA double-strand break (DSB) into a preselected genomic site and exploit the cellular Homology Directed Repair (HDR) pathway to insert a corrective sequence from an exogenous DNA template. This strategy has been widely used to insert a functional complementary DNA (cDNA) copy of the mutant gene downstream its endogenous promoter, thus restoring both its function and physiologic expression control and limiting the potential genotoxicity of the procedure to the nuclease on- and off-target genomic sites (Lombardo *et al*, 2007; Li *et al*, 2011; Barzel *et al*, 2015). Recent studies used this approach to integrate a wild-type *CD40LG* cDNA into the first exon of HIGM1 patient T cells (Hubbard *et al*, 2016) or HSPC (Kuo *et al*, 2018) and showed partial rescue of activation-dependent expression and functionality of the edited *CD40LG* gene. However, it remains unclear if a T-cell therapy can effectively correct the HIGM1 phenotype and if the low gene editing efficiency obtained in HSPC can be sufficient to rescue the disease. Moreover, the reported strategies failed to reconstitute full expression level of the edited gene and may not avoid uncontrolled CD40L expression from off-target vector integration.

Here, we developed a "one-size-fits-all" editing strategy for *CD40LG* correction conditional on on-target integration which fully reconstitutes CD40L expression in human healthy donors and HIGM1 patients' T cells and exploited recently improved protocols to apply it with high efficiency to T Stem Memory Cells (Cieri *et al*, 2013) and long-term repopulating HSPC (Schiroli *et al*, 2019; Ferrari *et al*, 2020). When directly applied to T cells, our strategy allows purifying corrected cells from unedited cells or those carrying genomic rearrangements at the target locus and eliminating engineered cells *in vivo* in case of adverse events. By exploiting the HIGM1 mouse model and wild-type murine cells as surrogate models of functional edited cells, we compared the therapeutic potential of adoptive T-cell transfer and HSPC transplantation with functional cell fractions that match current editing efficiencies. Both approaches showed substantial and durable benefits in restoring secondary humoral response and controlling a disease-relevant pulmonary infection, thus supporting the rationale for prioritizing first-in-human clinical testing in HIGM1 according to the relative risk–benefit.

## Results

### "One-size-fits-all" conditional correction strategy for safe rescue of physiological CD40L expression in human T cells

To correct the vast majority of HIGM1-causing mutations (~ 95%) (Lee *et al*, 2005), including deletions, with a single set of nuclease and donor HDR template, we designed a gene editing strategy aimed to insert a 5′-truncated corrective cDNA, which includes all downstream exons and the cognate 3′ UTR, within the first intron of the human *CD40LG* gene (Fig 1A). Differently from previously reported strategies (Hubbard *et al*, 2016; Kuo *et al*, 2018), the choice of targeting the first intron allows avoiding any promoter sequence or the full-length cDNA in the donor template, thus limiting potential ectopic/unregulated expression of the therapeutic cDNA in case of off-target integration of the donor and effectively making expression

conditional to targeted insertion in the intended locus. We screened a panel of *Streptococcus pyogenes* (*S.p.*) and *Staphylococcus aureus* (*S.a.*) Cas9 gRNAs targeting the *CD40LG* first intron (Fig EV1A) in primary human T cells by electroporating equal amounts of gRNA ribonucleoprotein complex (RNP) in the presence or absence of matching single-stranded donor oligonucleotides (ssODN) for targeted repair. Then, the two gRNAs showing the highest levels of on-target NHEJ-mediated indels and ssODN-mediated repair (Fig EV1B and C) were tested for HDR efficiency in combination with a cognate Adeno-Associated Vector 6 (AAV6) donor bearing a GFP reporter cassette. We selected for further development the *S.p.* g1 because it showed reproducibly higher levels of HDR integration in *CD40LG* of T cells (Fig EV1D). To stringently and comprehensively assess the off-target profile of g1, we coupled *in silico* prediction of closely matched sites in the reference human genome with two unbiased genome-wide analyses, GUIDE-seq (Tsai *et al*, 2015) and Digenome-Seq (Kim *et al*, 2015), and identified a total of 106 candidate off-target sites (OT, 58, 4, and 60, respectively) (Fig EV1E). Targeted resequencing of 93 OT candidates confirmed detectable nuclease activity on T cell only at one OT on chromosome 8 (Appendix Table S1). Despite this OT is not particularly dangerous, because located within an intergenic region distant 300 kb from the closest gene, by using a high-fidelity (HF) *S.p. Cas9* variant it was possible to reduce activity on this OT to undetectable levels in both T cells and $CD34^+$ HSPC, with no or limited impact on HDR efficiency, respectively (Fig EV1F and G). Thus, g1 was selected as leading gRNA for developing the *CD40LG* gene correction strategy.

We then optimized the donor template to achieve regulated and physiological expression of the edited *CD40LG* by integrating different templates configurations on T cells from male donors, which carry only one copy of the target gene, and using PMA/Ionomycin stimulation to induce CD40L expression. Proper and efficient splice trapping after targeted integration into *CD40LG* intron 1 was confirmed by testing different splice acceptor (SA) sequences followed by an in-frame *GFP* reporter (Fig EV1H). After editing, percentages of the resulting $GFP^+$ cells matched those of targeted integration measured by digital droplet PCR (ddPCR) on donor-genome junction (Fig EV1I) and all $GFP^+$ cells showed concurrent CD40L knockout (Fig EV1H), thus confirming that all the endogenous transcripts are efficiently spliced into the integrated cassette. When the *CD40LG* truncated cDNA followed by a GFP reporter cassette was used as donor template during editing, the $GFP^+$ cells showed regulated CD40L surface expression, which peaked at 6 h after PMA/Ionomycin stimulation and returned to baseline 1 day after (Fig EV1J; Van Kooten & Banchereau, 2000). However, the CD40L expression level in the edited cells was lower than that measured in untreated cells or their unedited ($GFP^-$) counterparts and was increased but without matching the unedited level after codon-usage optimization of the cDNA sequence (Fig EV1J). Inclusion in the donor template of different polyadenylation (polyA) signals (SV40- or CD40LG-derived), one or two short intervening introns from a naturally occurring gene, and a *CD40LG* enhancer previously reported downstream the endogenous 3′ UTR (Schubert *et al*, 2002) did not further improve expression of the edited gene (Fig EV1J). We thus selected the original codon-optimized *CD40LG* cDNA donor construct with the HBB SA site and the endogenous 3′ UTR and polyA sequences for further development.

## Efficient and functional *CD40LG* correction of human CD4 HIGM1 T cells

We produced an AAV6 delivery vehicle for the selected HDR donor template (Wang *et al*, 2016) (Fig 1B) and optimized RNP and AAV6 dose for maximizing editing efficiency (Fig EV1K–N). Using a previously developed culture protocol, based on CD3/CD28 stimulation in the presence of IL-7 and IL-15 cytokines (Cieri *et al*, 2013), which preserves T Stem Memory Cells (TSCM; Fig 1C), we reproducibly achieved ~ 35% on-target editing efficiency in bulk or $CD4^+$ purified T cells from healthy male donors, as measured by the percentage of $GFP^+$ cells (Figs 1D and EV1L) and the ddPCR-based molecular analysis (Figs 1E and EV1M). This human T-cell subset ($CD62L^+CD45RA^+$; Fig 1C) is endowed with long-term multipotent and self-renewal capacity (Gattinoni *et al*, 2011; Cieri *et al*, 2013), longevity, and robust potential for immune reconstitution, thus representing a clinically relevant T-cell subset in the context of adoptive T-cell therapies (Gattinoni *et al*, 2017). Importantly, the gene editing procedure similarly corrected TSCM and all the T-cell phenotypes analyzed (Central Memory CM, Effector Memory EM and Effector Memory RA TEMRA; Fig 1D), while preserving a relevant fraction of TSCM in culture (Fig 1F). Moreover, the cell manipulation did not skew TCR repertoire diversity if compared with untreated controls (Fig 1G).

We then challenged our *CD40LG* gene correction strategy with HIGM1 patient-derived CD4 T cells carrying a null c.334 G>T mutation in exon 3. We observed comparable or even higher gene correction efficiency in all T-cell subpopulations as compared to healthy donor cells (Fig 1D and E), maintaining TSCM subpopulation in culture and TCR repertoire diversity (Fig 1F and G). By assessing the expression of CD40L protein on the surface of patient-derived edited cells after PMA/Ionomycin stimulation, we found rescue of regulated expression at levels comparable to those of the edited healthy control, even if achieved in a lower fraction of cells (Fig 1H). Of note, a small fraction of edited $GFP^-$ (negative) patient T cells expressed CD40L on their surface (Fig 1H and I), suggesting the occurrence of HDR integrations which exploit the 3′UTR in the template as downstream homology, thus leaving out the GFP reporter cassette (Fig EV1O), as also previously reported by others (Hubbard *et al*, 2016). This explanation was confirmed by specific molecular analyses on sorted cells (Fig EV1O). Because expression of CD40L in $GFP^+$ and $GFP^-$ edited patient-derived T cells reached similar levels and regulation, we can exclude interference from the reporter expression cassette on the edited gene (Fig 1H). Notably, edited T cells from a HIGM1 patient carrying a rare c.135T>A mutation in exon 1, upstream of the insertion site of our donor template and thus not amenable to correction, showed no rescue of CD40L expression (Fig EV1P).

To test the functionality of the edited CD40L on treated CD4 T cells, we performed an *in vitro* class-switching assay (Fig 1J) and assessed by FACS analysis the ability of edited T cells to induce naive B-cell class-switch recombination (CSR) (Hubbard *et al*, 2016). Sorted $GFP^+$ T cells derived from healthy donor, co-cultured with allogeneic naive B cells (1:1 ratio), induced IgG class switching at rates comparable to those observed with unedited $GFP^-$ or untreated counterparts (Fig EV1Q). Of note, while HIGM1 patient-derived untreated cells failed to facilitate *in vitro* B-cell CSR, $GFP^+$ corrected T cells promoted class switching at similar levels as

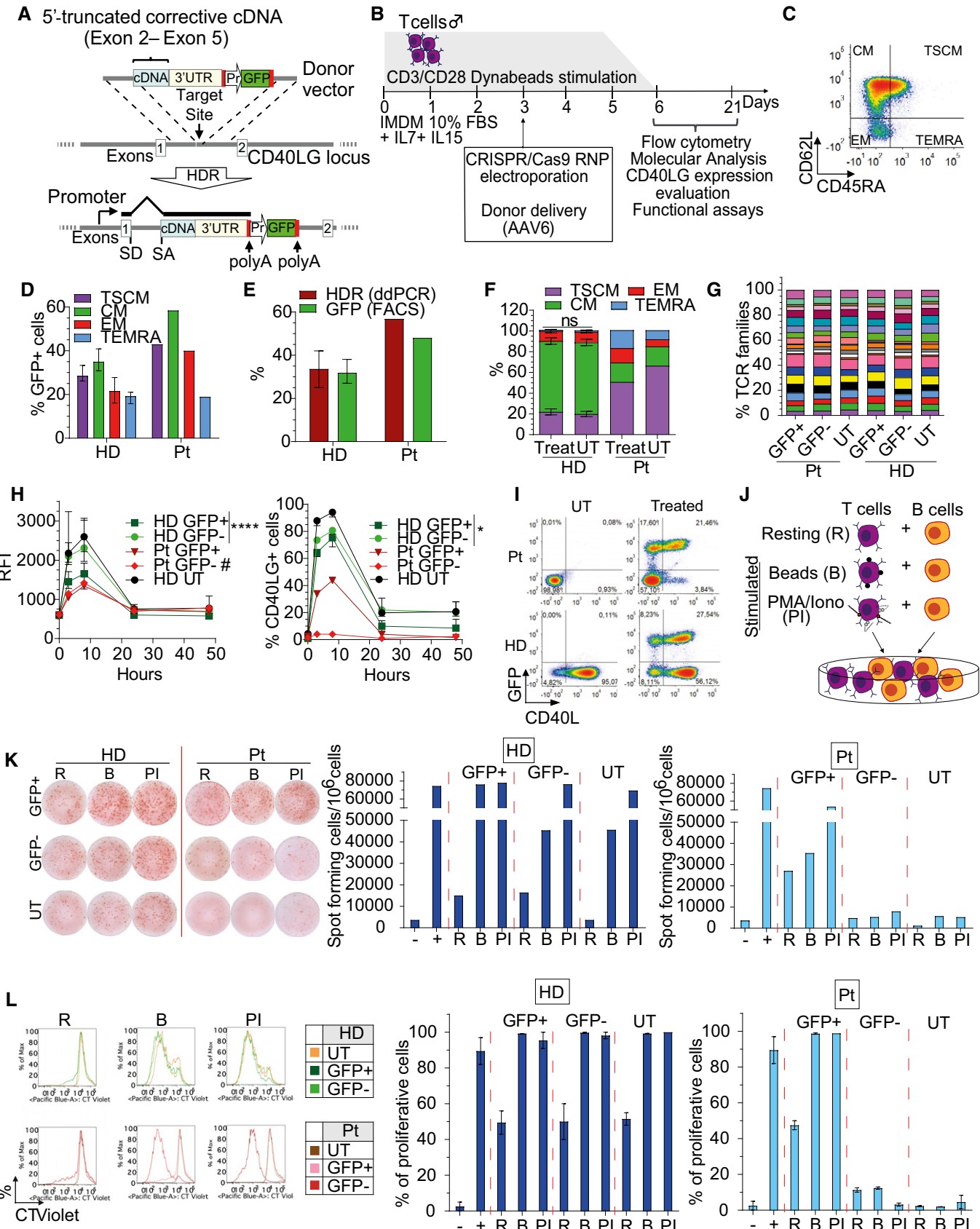

Figure 1.

◀

**Figure 1.  Efficient, safe, and functional *CD40LG* correction of human HIGM1 T cells by "One-size-fits-all" conditional correction strategy.**

A, B    Schematics of (A) gene editing strategy and donor DNA template for intron 1 *CD40LG* target locus and (B) experimental protocol in T cells.

C    Representative plot showing gating strategy for CD4$^+$ T-cell phenotypes. Effector Memory RA (TEMRA): CD45RA$^+$CD62L$^-$; Effector Memory (EM): CD45RA$^-$CD62L$^-$; Central Memory (CM): CD45RA$^-$CD62L$^+$; Stem Cell Memory (TSCM): CD45RA$^+$CD62L$^+$.

D    Percentage of GFP$^+$ cells within T-cell subpopulations 17 days after *CD40LG* editing of healthy male donor (HD; $n = 11$) or patient (Pt; $n = 1$) derived CD4$^+$ T cells, measured by FACS analysis. Median $\pm$ IQR.

E    Percentage of GFP$^+$ cells or HDR in T cells from (D). Median $\pm$ IQR.

F    Population composition in male HD or patient-derived untreated ($n = 9$ HD UT, 1 Pt UT) or bulk edited T cells (Treat) from (D). For each subpopulation, the comparisons between HD Treated vs. UT were performed by using LME models, with random effects defined to account for the same donor and for different number of replicates per donor within group. All *P*-values were adjusted with Bonferroni's correction to account for multiple testing. In all LME models, the percentages were used in square root scale to meet the assumption of normality of the residuals of the model. Mean $\pm$ SEM.

G    Percentage of TCRBV families detected by spectratyping. UT, sorted edited (GFP$^+$), sorted non-edited (GFP$^-$) CD4$^+$ T cells derived from male HD ($n = 1$) or patient (Pt; $n = 1$) were analyzed at 17 days after *CD40LG* editing.

H    Time course of CD40L surface expression after PMA/Ionomycin stimulation measured by Relative Fluorescence Intensity (RFI, normalized to T0; left) and percentage (right) on UT ($n = 3$), edited (GFP$^+$) or unedited (GFP$^-$) HD or Pt CD4$^+$ T cells from (D) ($n = 9$ HD, 1 Pt). Longitudinal comparisons between HD GFP$^+$ vs. GFP$^-$ were performed with an LME model, accounting for multiple donors and separately for RFI and %CD40L$^+$ cells (see Appendix Supplementary Statistical Methods). The reported statistical comparisons refer only to 8 h time-point (****$P < 0.0001$ and *$P = 0.0450$, respectively). #measured on the small fraction of CD40L$^+$ cells. Median $\pm$ IQR.

I    Representative plots showing CD40L and GFP expression in UT or bulk edited (Treated) CD4$^+$ T cells derived from male HD or Pt from (D) at 8 h after PMA/Ionomycin stimulation.

J    Cartoon depicting protocol of B–T-cell co-culture used to assess functionality of corrected T cells. CD4$^+$ T cells were stimulated with beads (B) or with PMA/Ionomycin (PI) or kept resting (R) prior to co-culture with allogeneic naive B cells (1:1 ratio). B cells cultured alone or in presence of soluble CD40L (sCD40L) were used as negative and positive controls, respectively.

K    Left: IgG positive spots resulting from B–T-cell co-culture. Right: IgG$^+$ secreting B cells, evaluated by ELISPOT assay. B cells were isolated from peripheral blood (PB) of HD and co-cultured with male HD or Pt sorted GFP$^+$, GFP$^-$ and UT T cells, resting (R) or stimulated with beads (B) or PMA/Ionomycin (PI). B cells cultured alone ($-$) or in presence of sCD40L ($+$) were used as negative and positive controls, respectively ($n = 1$ for each group).

L    Left: Histograms representing proliferation results from B–T-cell co-culture. Right: Analysis of B-cell proliferative capacity by Cell Trace dilution assay in allogeneic sorted B cells isolated from PB of HD and co-cultured with HD or Pt T cells from (K). B cells cultured alone ($-$) or in presence of sCD40L ($+$) were used as negative and positive controls, respectively ($n = 2$ for each group). Mean $\pm$ SEM.

Source data are available online for this figure.

healthy control (Fig EV1Q). We also observed relatively high levels of CSR in the sorted GFP$^-$ fraction of patient-derived cells (Fig EV1Q), likely reflecting the presence of some edited cells, as shown above, and indicating that even few functional T cells can turn many B cells positive for IgG immunostaining in this assay. Thus, we improved the stringency of the analysis by evaluating IgG secretion by ELISPOT assay and B-cell proliferation. These experiments showed that corrected GFP$^+$ CD4 T cells derived from both healthy donor and patient were able to induce IgG secretion and B cells proliferation at levels similar to unedited and untreated healthy controls, while untreated or sorted GFP- patient cells failed to do so (Fig 1K and L).

Overall, these data show that our gene editing strategy efficiently rescues regulated expression and functional activity of CD40L on HIGM1 patient-derived CD4 T cells.

**Adding a selector to the therapeutic gene allows enriching for edited cells and rescuing physiological CD40L expression level**

In order to increase the total amount and relative proportion of functional cells in the therapeutic product, we aimed to enrich for corrected cells *in vitro* by transcriptionally coupling the corrective cDNA to a clinically compatible selector (C-terminal truncated low-affinity NGFR receptor, hereafter named NGFR) by an internal ribosome entry site (IRES) sequence (Fig 2A). Despite the increased size of the donor template, we reproducibly obtained on-target editing efficiencies in CD4 healthy donor and HIGM1 patient (c.242_243insT mutation in exon 2)-derived T cells comparable to those shown above within all the T-cell subpopulations for the *CD40LG* cDNA alone (~ 35%; Figs 2B, bulk and EV2A), including

TSCM stringently defined as CD95$^+$CCR7$^+$CD45RO$^+$CD62L$^+$CD45RA$^+$ (Fig EV2B and C; Cieri *et al*, 2013). Intriguingly, the new construct allowed full rescue of the edited *CD40LG* gene expression to levels matching that of unedited controls in both healthy donor and patient-derived cells, while preserving regulated expression dependent on PMA/Ionomycin-mediated activation (Figs 2C and D, and EV2D). Consistent with CD40L restoration, we confirmed the ability of corrected patient-derived T cells to induce IgG secretion by co-cultured B cells, as opposed to unedited cells (Fig EV2E). Surprisingly, although CD40L protein was detectable on the surface of edited T cells only after activation, NGFR was detectable even in basal conditions (Fig 2D). This finding could be explained by differential sorting of the CD40L and NGFR proteins into the regulated vs. constitutive cell secretory pathway (Casamayor-Palleja *et al*, 1995; Koguchi *et al*, 2007), respectively. We confirmed this hypothesis by performing intracellular co-staining of CD40L and NGFR (Fig EV2F) and showing that CD40L was accumulated within intracellular stores while NGFR was expressed on the surface of edited cells (Fig EV2G). Because upon cell activation CD40L is translocated to the membrane by regulated secretion, its surface expression level might be restored to physiological levels once the stores have been replenished above a certain threshold. We thus took advantage of the surface expression of NGFR in basal conditions to sort edited cells without relying on additional stimulation, which would accelerate cellular exhaustion. By exploiting immunomagnetic separation with NGFR-specific beads, we enriched edited cells up to 80% with an average loss of 10% of edited cells in the NGFR$^-$ fraction (Fig 2B). Moreover, the sorting procedure did not affect culture composition, preserving the TSCM subset *in vitro* (Fig 2E). In order to assess the impact of the entire

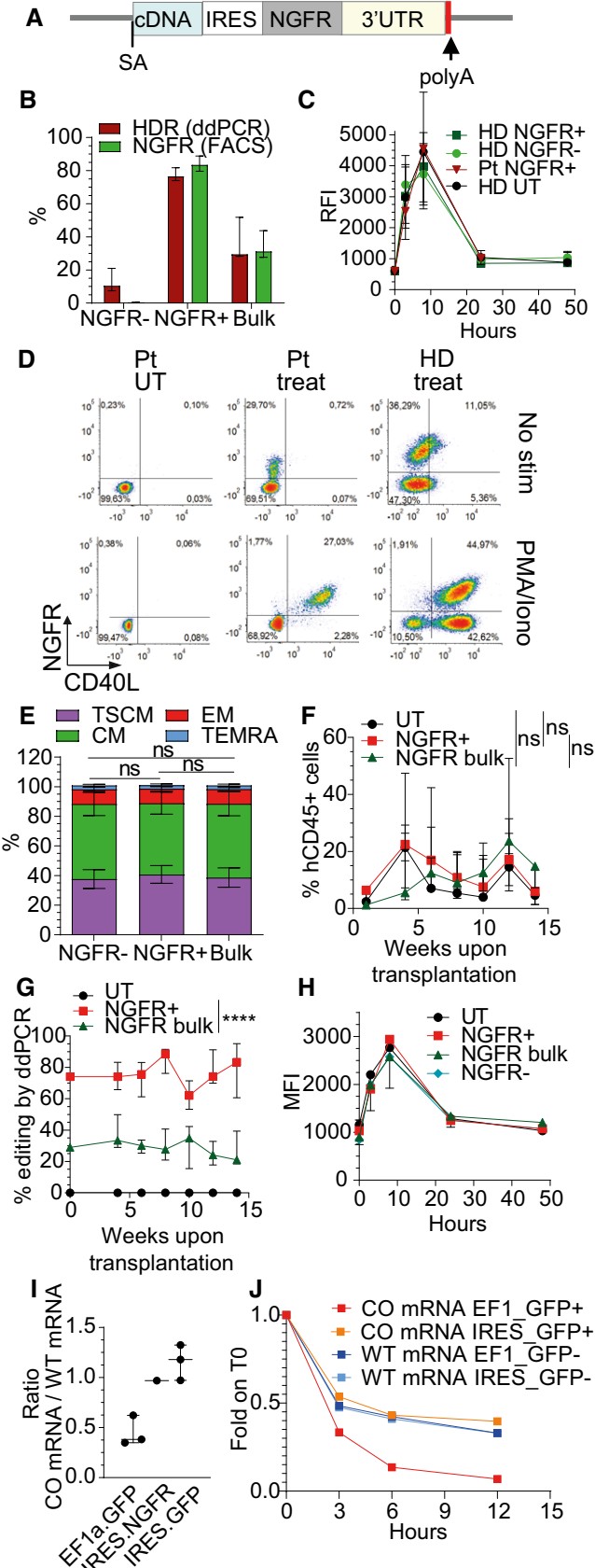

**Figure 2. Edited cells enrichment and increased CD40L expression level by adding NGFR selector to the therapeutic gene.**

A  Schematics of donor DNA template used for edited cells enrichment.

B  Percentage of NGFR+ cells or HDR at 17 days after *CD40LG* editing of male HD bulk edited, sorted NGFR+ or sorted NGFR− CD4+ T cells (*n* = 7 for each group). Median ± IQR.

C  Time course of CD40L surface expression after PMA/Ionomycin stimulation measured by RFI (normalized to T0) on UT (*n* = 2), edited (NGFR+) or unedited (NGFR−) HD or Pt derived CD4+ T cells (*n* = 4 HD, 1 Pt). Median ± IQR.

D  Representative plots showing CD40L and NGFR expression in UT or bulk edited (Treated) CD4+ T cells derived from male HD or Pt from (C) before and 8 h after PMA/Ionomycin stimulation.

E  Population composition in male HD bulk edited, sorted NGFR+ or sorted NGFR−CD4+ T cells from (B) (*n* = 7 for each group). Friedman test (to account for the same donor) with *P*-values adjusted with Bonferroni's correction to account for multiple testing. Mean ± SEM.

F  Human CD45+ cell engraftment in PB after transplantation of male HD UT (*n* = 5), bulk edited (NGFR bulk; *n* = 5) or sorted NGFR+ (*n* = 5) CD4+ T cells. Longitudinal comparisons were performed with an LME model, followed by an appropriate *post hoc* analysis (see Appendix Supplementary Statistical Methods). The reported statistical comparisons refer only to the overall difference among groups. Median ± IQR.

G  Percentage of HDR within human cells over time, measured by ddPCR in PB of mice from (F) (*n* = 5 for each group, measured in engrafted mice with sufficient blood material). Longitudinal comparisons were performed with an LME model with time treated as a continuous variable (see Appendix Supplementary Statistical Methods). UT group was not included in the analysis. Effect of the time and the eventual different effect of the time in the groups were not retained in the final model after backward variable selection, highlighting a constant (and significantly different, ****P < 0.0001) behavior of the groups over time. Median ± IQR.

H  Time course of CD40L surface expression after PMA/Ionomycin stimulation measured by MFI on pooled CD4+ T cells retrieved from spleens of mice from (F) (*n* = 1 for each group, except for NGFR− unedited group, *n* = 3). Median ± IQR.

I  Dot plot depicting the ratio between expression of *CD40LG* edited mRNA (codon-usage optimized, CO) and *CD40LG* wild-type mRNA (WT) in cells edited with three different donor templates from Fig EV2H. Three independent experiments (*n* = 3 EF1a.GFP, 1 IRES.NGFR, 3 IRES.GFP). Median ± IQR.

J  Time course of expression of WT-*CD40LG* mRNA or CO-*CD40LG* mRNA, measured as fold change (FC) on IPO8 housekeeping gene and normalized to T0. Sorted edited (+) and sorted unedited (−) CD4+ T cells were analyzed before and 3, 6, and 12 h after Actinomycin D treatment (*n* = 1 for each group).

Source data are available online for this figure.

manipulation procedure and rule out selection of CD4 T cells with limited fitness, we evaluated engraftment and survival of the edited cells *in vivo*, by performing adoptive transfer of male bulk edited T cells, NGFR+ enriched T cells or untreated cells into non-obese diabetic (NOD)-severe combined immunodeficiency (SCID)-IL2Rg−/− (NSG) mice. We observed similar T-cell engraftment kinetics and persistence across all groups of mice (Fig 2F) and, importantly, editing efficiencies of transferred cells remained stable from pre- to post-transfer, reaching up to 80% in the mice transplanted with NGFR-enriched cells (Fig 2G). Of note, edited CD4 T cells retrieved from spleens of transplanted mice retained their ability to express surface CD40L with regulated kinetics upon stimulation with PMA/Ionomycin (Fig 2H).

To further investigate the mechanism underlying the full rescue of CD40L expression in T cells edited with the new corrective template, we tested whether (i) IRES sequence *per se*, (ii) the

presence of a coding sequence downstream the IRES, or (iii) the length of the transcript *per se* could influence CD40L accumulation and thus its surface expression. We performed targeted integration in CD4 T cells of new donor constructs carrying the corrective *CD40LG* cDNA coupled to (i) IRES sequence alone, (ii) IRES followed by a sequence encoding for a reporter gene other than NGFR (CD19 or GFP), (iii) IRES coupled to a non-coding sequence, or engineered with two long intervening introns from a naturally occurring gene (Fig EV2H). We found full restoration of CD40L surface expression only in the presence of a coding sequence downstream the IRES (Fig EV2I). To investigate the molecular mechanism underlying this rescue, we measured mRNA expression of the WT and edited *CD40LG* gene with different constructs in sorted edited cells by ddPCR assays specific either for the wild-type or codon-usage optimized mRNA exons-junctions (Fig EV2J). We found that *CD40LG* mRNA levels before and after PMA/Ionomycin stimulation were substantially lower for the edited allele unless the donor construct comprised an IRES-coding sequence, which nearly matched the unedited controls (Figs 2I and EV2K), thus corresponding to the different protein expression levels measured. We then assessed the stability of the different mRNAs by chasing the *CD40LG* mRNA level after blocking transcription by Actinomycin D in PMA/Ionomycin sorted stimulated cells (Fig EV2J). Intriguingly, the edited transcript had a substantially faster decay than the unedited one, whereas addition of an IRES-coding sequence rescued its stability to the unedited control level (Fig 2J).

Overall, the adopted selection strategy based on an IRES-comprising bicistronic transcript expressing the corrective *CD40LG* cDNA and a selector unexpectedly allowed both enrichment of edited cells in basal conditions and full rescue of regulated CD40L expression on the cell membrane.

## Specific depletion of edited cells by exploiting a modified clinically compliant selector

Since we found that the selector gene is expressed also in unstimulated edited T cells, we envisioned using this surface tag to deplete them in case of possible adverse events. Nevertheless, the NGFR selector gene is not a suitable candidate for antibody-mediated depletion, since it is expressed by a variety of tissues. We thus reasoned that a truncated version of human EGF receptor (hEGFRt) (Wang *et al*, 2011) could better fit our purpose, since it is recognized by pharmaceutical grade anti-EGFR monoclonal antibody (Cetuximab), which displays manageable and dose-related side effects in the clinic. hEGFRt is devoid of cytoplasmic and extracellular N-terminal ligand binding domains, while retaining the transmembrane domain and an intact binding epitope for Cetuximab (Wang *et al*, 2011). We thus cloned a new corrective construct by inserting the reported sequence (Wang *et al*, 2011) of hEGFRt downstream to the IRES (Fig 3A) and performed targeted integration experiments in order to assess its expression on the surface of edited T cells. Unfortunately, no hEGFRt protein was detectable on T-cell membrane in basal conditions, becoming measurable only after PMA/Ionomycin stimulation (Fig 3B). We hypothesized that either the absence of a cytoplasmic domain might reduce the stability of the protein or the heterologous signal peptide used in the published construct might not efficiently guide the protein to the membrane. We thus replaced the signal peptide with that of NGFR and fused downstream the

transmembrane domain a short cytoplasmic tail, either derived from the NGFR cytoplasmic domain (named EGFRmod1) or comprising the first eight amino acids of EGFR intracellular domain, not involved in the tyrosine kinase activity (named EGFRmod2) (Fig 3A). By editing the cells using these modified donor templates, hEGFRt protein was expressed on the surface of edited T cells even in the absence of stimulation (Fig 3B) and, importantly, we observed similar gene editing efficiency (Fig EV3A), culture composition (Fig EV3B), CD40L regulated expression (Fig EV3C), and functionality (Fig EV3D and E) as compared to previous templates. *In vivo* depletion of hEGFRt-expressing cells by Cetuximab relies on antibody-dependent cellular cytotoxicity (ADCC), which also requires functional NK cells (Lee *et al*, 2011). Since ADCC on human cells is difficult to be assessed in xenotransplantation experiments with immunodeficient mice, we explored an *in vitro* immuno-toxin-based strategy to evaluate whether edited cells carrying hEGFRt were amenable to antibody-mediated depletion (Palchaud-huri *et al*, 2016). By culturing edited T cells in the presence of Cetuximab conjugated to the protein synthesis inhibitor toxin saporin (Cetuximab-SAP) or of antibody and toxin alone as controls, we observed substantial depletion (~ 50%) of hEGFRt-expressing lymphocytes at both doses tested (Fig 3C and D). While the decreased internalization rate of our modified hEGFRt is likely reducing the efficacy of immunotoxin treatment, these data suggest that hEGFRt is a suitable candidate both for *in vitro* selection and *in vivo* depletion of *CD40LG* edited cells.

## Editing *CD40LG* in long-term repopulating human HSPC

Since the reconstituted repertoire and chimerism of edited T cells might be limiting in the long term and their transfer alone cannot correct all affected hematopoietic lineages, we investigated application of our editing strategy to HSPC. We thus performed *CD40LG* gene editing in human CD34[+] male HSPC (Fig 4A), using the same reagents as above in the context of recently optimized protocols from our laboratory (Schiroli *et al*, 2019; Ferrari *et al*, 2020). When using a donor template comprising only the corrective cDNA, we achieved up to 40% targeted integration in the most primitive subpopulation (CD34[+]CD133[+]CD90[+], Fig 4B), while preserving the culture composition after the manipulation (Fig 4C). Longer templates carrying the selector cassette yielded lower editing efficiency in the more primitive cells (Fig 4B), possibly due to poor permissiveness of these cells to editing manipulation.

We recently reported that transient p53 inhibition during HSPC editing increases the yield of clonogenic and repopulating HSPC and the size and clonality of hematopoietic reconstitution (Schiroli *et al*, 2019; Ferrari *et al*, 2020). Thus, we co-electroporated an mRNA encoding for a dominant negative p53 truncated form (GSE56) when editing *CD40LG* locus and transplanted the treated cells into NSG mice (Fig 4A). We confirmed that cells transiently treated with the p53 inhibitor engrafted faster and to higher levels as compared to cells edited without it (Fig 4D). Moreover, gene editing efficiency remained high in long-term grafts of HSPC edited with GSE56 (up to 30% in peripheral blood (PB)) whereas they decreased progressively with time in grafts of HSPC edited without it, suggesting better preserved contribution of edited long-term repopulating cells in the former condition (Fig 4E). Targeted integration levels in hematopoietic organs and sorted myeloid, lymphoid, and progenitor

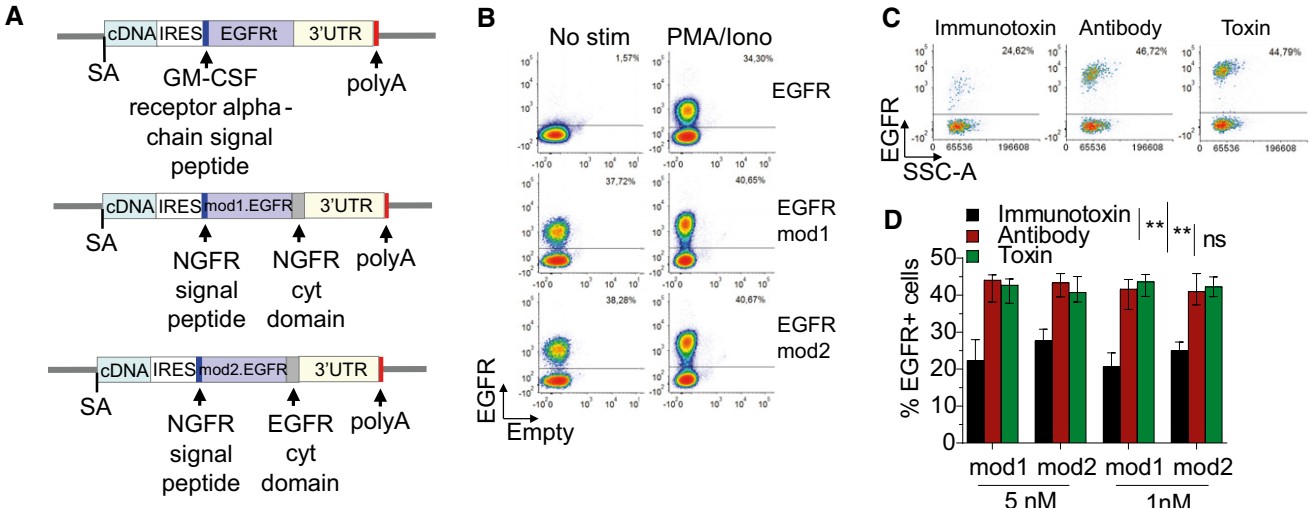

**Figure 3. Selection and depletion of the edited T cells by exploiting an optimized hEGFRt gene.**

A  Schematics of donor DNA templates carrying corrective cDNA coupled to IRES sequence and hEGFRt selector genes.
B  Representative plots showing hEGFRt expression in bulk edited CD4[+] T cells derived from male HD before and 8 h after PMA/Ionomycin stimulation. Cells were edited with the three constructs depicted in (A).
C  Representative plots showing EGFR expression in bulk edited CD4[+] T cells derived from male HD at 3 days after treatment with immunotoxin (left), antibody (middle), or toxin (right).
D  Bar plot showing percentage of EGFR[+] T cells at 3 days after treatment with 5 nM or 1 nM of immunotoxin, antibody, or toxin, measured by FACS Analysis. Friedman test with Dunn's multiple comparisons. P-values were adjusted with Bonferroni's correction to account for multiple comparisons (**P = 0.0052 and **P = 0.0024 for immunotoxin vs. antibody and **P = 0.0010 and **P = 0.0024 for immunotoxin vs. toxin, referring to EGFRmod1 and EGFRmod2, respectively). Different dose conditions were used as a unified group for statistical analysis (n = 10 for each group). Median ± IQR.

Source data are available online for this figure.

cell populations from the bone marrow were consistent with those measured in PB cells at the end of the experiment for all groups (Fig 4F and G). Importantly, edited CD4 T cells retrieved from the spleen of transplanted mice retained their ability to express surface CD40L after stimulation with PMA/Ionomycin (Fig 4H).

Overall, these data show effective editing of the *CD40LG* gene in human long-term repopulating HSPC and regulated expression of the edited allele in their T-cell progeny.

**Adoptive transfer of functional T cells partially rescues IgG response in *Cd40lg*[−/−] mice**

To investigate and compare the therapeutic benefit of adoptive T-cell transfer and HSPC transplantation therapies with autologous gene corrected cells, we took advantage of *Cd40lg*[−/−] mice, which faithfully recapitulate the human HIGM1 phenotype (Renshaw *et al*, 1994). Since subjects with HIGM1 mutations have preserved cytotoxic T-cell immunity, we first tested whether the transfer of CD40L-expressing T cells might lead to undesirable immunogenicity due to the presentation of epitopes never experienced before by the host immune system. Thus, we transplanted two 10-fold different doses (2 × 10[6] and 20 × 10[6]) of syngeneic wild-type (WT) CD3 T cells into *Cd40lg*[−/−] or WT mice and compared their engraftment and persistence in the two hosts. We found comparable levels of engraftment over time in the PB of both types of recipients at each input cell dose and, after 1 month from the transplant, in secondary lymphoid organs (Fig EV4A and B), indicating that CD40L-expressing T cells

are not rejected in a *Cd40lg*[−/−] host even in the absence of immunosuppression.

We then evaluated the impact of a pre-transplant conditioning regimen on the engraftment level of adoptively transferred T cells and infused 10[7] WT CD3 T cells into *Cd40lg*[−/−] mice after treatment or not with a lymphodepleting chemotherapeutic agent (cyclophosphamide, CPA; Fig 5A). CPA treatment induced a transient leucopenia in the recipient (Fig 5B), which resulted in a threefold higher level of donor T-cell engraftment in PB (Fig 5C) as compared with not conditioned mice. Donor T cells showed an early wave of expansion followed by contraction in the pre-conditioned mice and were then maintained in PB of both groups to stable sustained levels until the end of the experiment at 6 months post-injection. Engraftment levels in the secondary lymphoid organs between experimental groups corresponded to those observed in PB (Fig EV4C). We observed greater expansion of CD8 over CD4 donor T cells in PB and organs of mice treated with CPA as compared with the untreated group (Figs 5D and E, and EV4D and E). Such behavior was not observed in the recipient T cells, irrespective of conditioning or not (Fig EV4F–I), suggesting a growth advantage of the transplanted CD8 T cells during the homeostatic proliferation triggered by conditioning.

Similar to the effects of *CD40LG* mutations in humans, *Cd40lg*[−/−] mice fail to mount secondary antigen-specific responses to immunization with thymus-dependent antigens (Xu *et al*, 1994). To assess whether adoptive transfer of WT T cells can restore immunoglobulin class switching, we vaccinated the transplanted mice with

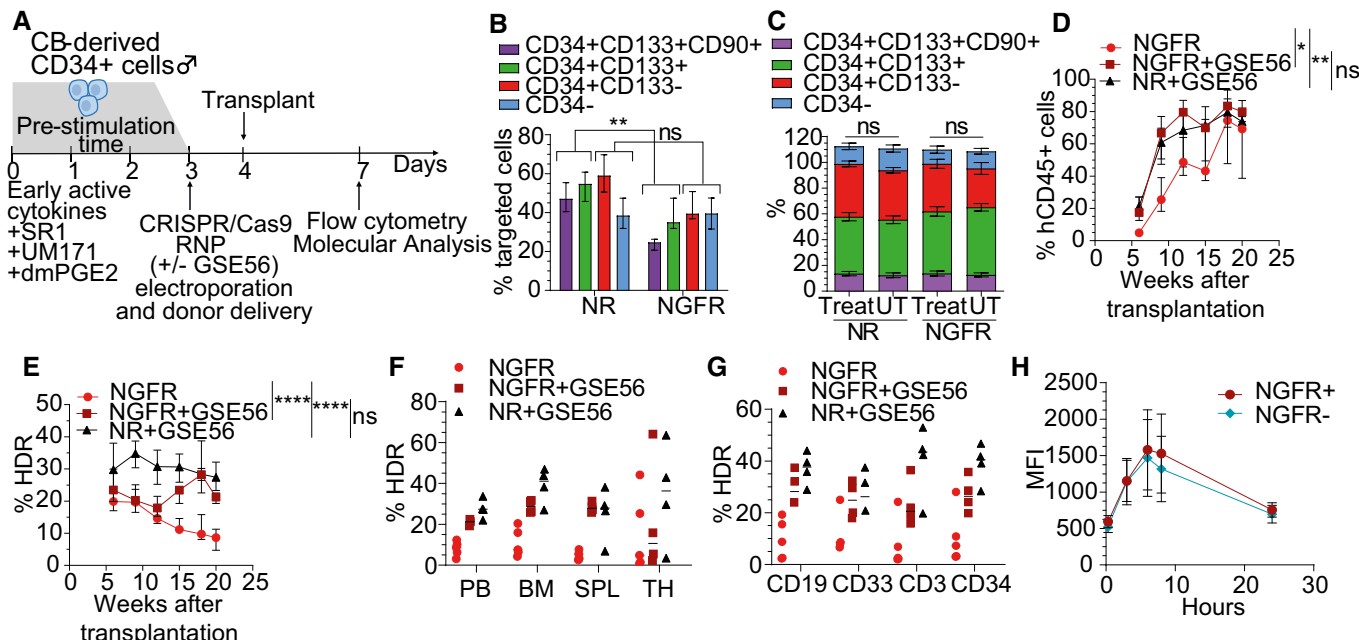

**Figure 4. CD40LG gene editing in long-term repopulating human HSPC.**

A    Schematics of experimental protocol in HSPC.

B    Percentage of HDR in HSPC subpopulations at 4 days after *CD40LG* editing, measured by molecular analysis (ddPCR). Cells were edited with donor template either comprising only the corrective cDNA (NR; n = 4), or carrying the selector cassette IRES.NGFR (NGFR; n = 5) and sorted according to surface markers: CD34⁻; CD34⁺CD133⁻CD90⁻; CD34⁺CD133⁺CD90⁻; CD34⁺CD133⁺CD90⁺. CD34⁺CD133⁺CD90⁺/CD90⁻ and CD34⁺CD133⁻/CD34⁻ conditions were used as unified groups for statistical analysis. The comparisons between the groups were performed with LME models accounting for the different subpopulations included in the analysis and with random effects defined to account for the same donor. In comparison about CD34⁺CD133⁺CD90⁺/CD90⁻, the percentages were used in square root scale to meet the assumption of normality of the residuals of the model. For CD34⁺CD133⁺CD90⁺/CD90⁻, **P = 0.0035, for CD34⁺CD133⁻/CD34⁻ P = 0.2051. Median ± IQR.

C    Population composition in UT or bulk edited HSPC from (B) (n = 8 for each group, except UT cells from NGFR group, n = 7). Paired Wilcoxon's test, to account for the same donor, with P-values adjusted with Bonferroni's correction to account for multiple testing. Analysis was performed separately for NR and NGFR groups. Mean ± SEM.

D    Human CD45⁺ cell engraftment in PB after transplantation of HSPC edited with the two donor templates described in (B) and with (n = 4) or without (n = 5) GSE56. Comparison between groups was performed by NLME model with an asymptotic model (see Appendix Supplementary Statistical Methods). The reported statistical comparisons refer only to the first time-point (i.e., 6 weeks; *P = 0.0213 and **P = 0.0069). Group transplanted with HSPC treated without GSE56 needs significantly greater time for reaching the final plateau with respect to both GSE56 groups (P = 0.0034 for NGFR⁺GSE56 and P = 0.0080 for NR⁺GSE56). Median ± IQR.

E    Percentage of HDR within human cells over time, measured by ddPCR in PB of mice from (D). Longitudinal comparisons were performed with an LME model with time treated as a continuous variable, and followed by an appropriate *post hoc* analysis (see Appendix Supplementary Statistical Methods). The reported statistical comparisons refer only to the last time-point (i.e., 20 weeks; ****P < 0.0001 in both comparisons). NR⁺ GSE56 group shows a constant behavior over time (P = 0.0588). The slope of the linear trajectory over time is not significantly different between GSE56 groups (P = 0.1624), indicating gene editing stability of both groups over time. Mice transplanted with HSPC treated without GSE56 shows significantly lower slope than GSE56 groups (P < 0.0001 in respect of NGFR group, P = 0.0054 in respect of NR group), thus highlighting decreased gene editing efficiency over time. Median ± IQR.

F, G    Percentage of HDR within human cells in (F) PB, bone marrow, spleen, thymus and (G) within BM-sorted human subpopulations (CD19⁺, CD33⁺, CD3⁺, CD34⁺), measured by ddPCR in mice from (D). Median.

H    Time course of CD40L surface expression after PMA/Ionomycin stimulation measured by MFI on pooled CD4⁺ T cells retrieved from spleen of mice from (D) (n = 4 NGFR⁺, 4 NGFR⁻). Mean ± SEM.

Source data are available online for this figure.

Trinitrophenyl-conjugated Keyhole Limpet Hemocyanin (TNP-KLH) and measured serum levels of anti-TNP-KLH IgGs before boosting (PRE), performed 2 weeks after primary vaccination and 7 days after the boost (POST). A recall challenge was given after 3–4 months and TNP-KLH-specific IgGs were measured the day before (PRE) and 10 days after the second boost (POST) (Fig 5A).

Whereas non-transplanted *Cd40lg*⁻/⁻ mice produced nearly undetectable amounts of anti-TNP-KLH IgGs, the groups of mice transplanted with WT T cells showed a significant rescue in switched-antibody responses to single and recall vaccinations, albeit reaching levels below those of control WT mice (Fig 5F). Higher and stable (166 days from primary immunization) levels of TNP-KLH-specific IgGs and mild boosting responses to secondary immunization were detected only in mice transplanted after CPA conditioning (POST, Fig 5F). At the end of the experiment, spleens were harvested and analyzed by immunostaining with Anti-Peanut Agglutinin (PNA), which binds to PNA-reactive glycans expressed by germinal center (GC) B cells (Fig EV4J). Whereas non-transplanted

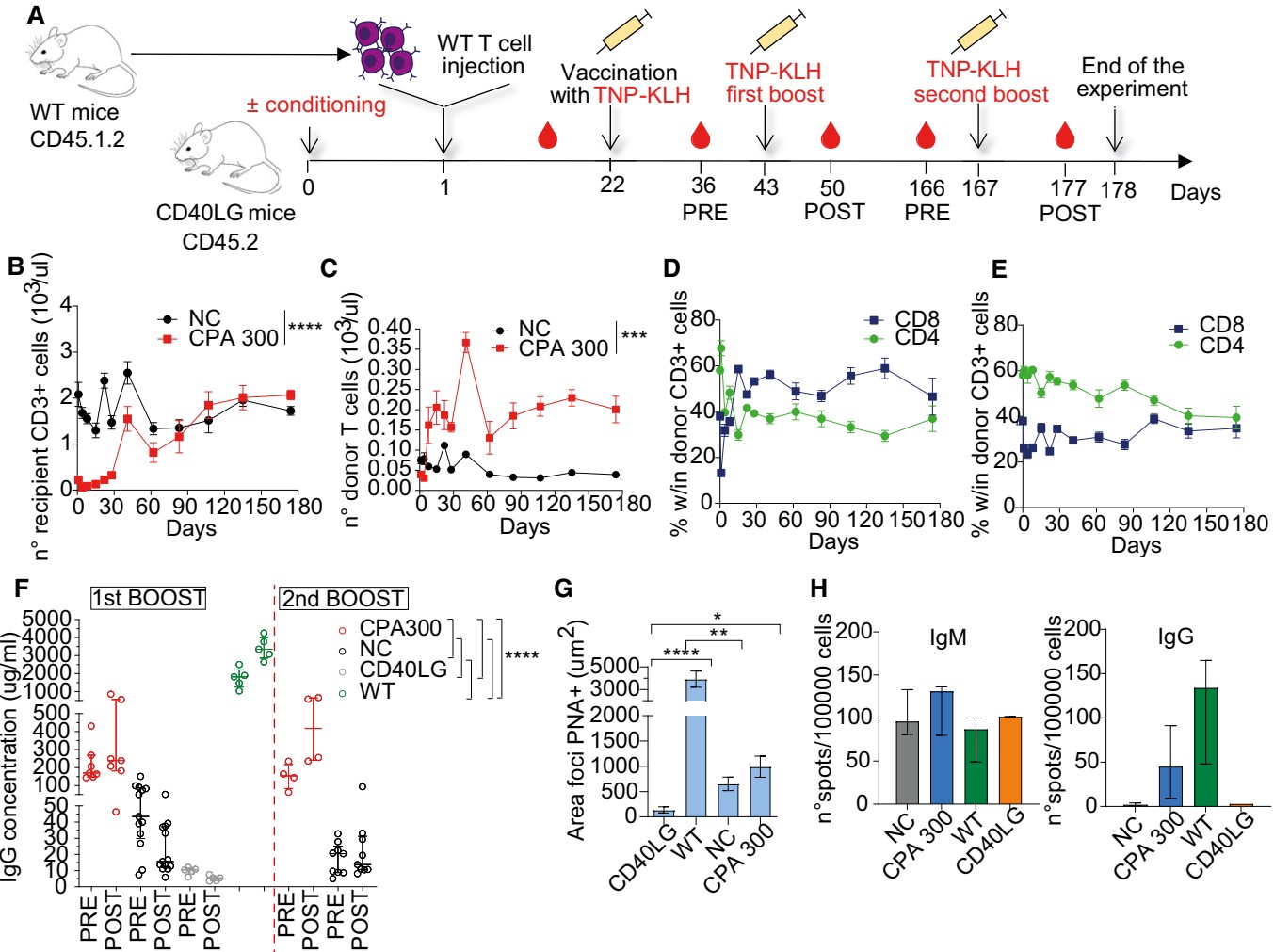

**Figure 5. Partial rescue of the IgG response by adoptive transfer of naive functional T cells in *Cd40lg*[−/−] mice.**

A   Schematics of adoptive transfer of spleen derived CD45.1.2[+]CD3[+] WT T cells (donor) into CD45.2 *Cd40lg*[−/−] mice (recipient) pre-conditioned with 300 mg/kg CPA or not (NC) and immune response studies.

B   Total counts of recipient CD3[+] cells gated within CD45.2[+] cells in PB of experimental mice (*n* = 4 CPA 300, 9 NC). Comparison between groups was performed by NLME model with an asymptotic model (see Appendix Supplementary Statistical Methods). The reported statistical comparisons refer only to day 1 (****$P$ < 0.0001). Mean ± SEM.

C   Total counts of engrafted donor CD45.1.2[+] cells in PB of mice from (B). Comparison between groups was performed by NLME model with an asymptotic model (see Appendix Supplementary Statistical Methods). The reported statistical comparisons refer only to plateau (***$P$ = 0.0001). Mean ± SEM.

D, E   Percentage of donor CD4[+] and CD8[+] cells gated within CD3[+]CD45.1.2[+] cells in PB of CPA treated (D) or NC mice (E) from (B). Mean ± SEM.

F   TNP-KLH-specific IgG in sera of transplanted mice collected at the times indicated in (A) (*n* = 7 CPA 300, 13 NC for first boost and 4 CPA 300, 9 NC for second boost). Sera from vaccinated WT (*n* = 5) and *Cd40lg*[−/−] (*n* = 5) mice were used as positive and negative controls, respectively. Two independent experiments. Comparisons of first boost data were performed with an LME model, accounting for multiple experiments, followed by appropriate *post hoc* analysis (see Appendix Supplementary Statistical Methods). The reported statistical comparisons refer only to the overall difference among groups (****$P$ < 0.0001 in all comparisons). Median ± IQR.

G   Average area of PNA[+] foci in splenic sections of experimental mice from (B), calculated as ratio between total PNA[+] area and number of PNA[+] foci/cells. Kruskal–Wallis test followed by *post hoc* analysis with Dunn's test (*n* = 9 CD40LG, 6 WT, 18 NC, 8 CPA300). *P*-values were adjusted with Bonferroni's correction to account for multiple comparisons (*$P$ = 0.0149, **$P$ = 0.0065 and ****$P$ < 0.0001). Mean ± SEM.

H   Detection of splenic TNP-KLH-specific IgM (left) and IgG (right)-secreting cells by ELISPOT assay (*n* = 7 NC, 4 CPA300, 3 WT, 2 CD40LG). Spots were counted by an ELISPOT Reader using a size range of 0.005–1 mm. Median ± IQR.

Source data are available online for this figure.

*Cd40lg*[−/−] mice did not show clusters of PNA[+] cells resembling GC, all transplanted mice showed some clusters of PNA[+] cells albeit of much smaller area than those found in WT mice (Figs 5G and EV4J), indicating partial rescue of GC formation after

transplantation. By measuring the presence of TNP-KLH cells within the splenic B-cell compartment of the vaccinated mice, we found similar numbers of IgM spot-forming cells in all groups of mice, while IgG spot-forming cells were detected only in the WT and in

CPA conditioned groups (Fig 5H). By comparing the percentages and absolute numbers of engrafted CD4 or CD8 T cells to TNP-KLH-specific IgG concentrations in sera, we confirmed a positive correlation between T-cell engraftment and the secondary response to TNP-KLH (Fig EV4K and L).

Overall, these findings show a corrective potential of adoptive T-cell transfer in HIGM1 mice which depends on the engrafted T-cell dose and requires a conditioning regimen for robust and long-term rescue of the humoral response.

### Ex vivo culture does not negatively affect CD4 T-cell engraftment and function

In a therapeutic setting, autologous T cells will need to be cultured and activated *in vitro* to allow efficient genetic engineering, as described above (Lombardo *et al*, 2011). Thus, we investigated the efficacy of adoptive T-cell therapy in HIGM1 mice after *in vitro* stimulation and expansion of WT T cells with anti-CD3/CD28 beads (Fig 6A). Since CD4 T cells are the major effectors of CD40L signaling and can be outgrown *in vitro* (Foulds *et al*, 2002) and in recipient mice by their CD8 counterparts (Fig 5D), we stimulated only purified CD4 T cells and cultured them for 7 days. We then injected $10^7$ outgrown cells/mouse, representing the upper range of potential clinical doses of cells/kg, in HIGM1 mice conditioned either with two different doses of CPA, anti-lymphocyte serum (ALS), or an anti-CD4 antibody (Fig 6A). Longitudinal analysis of recipient CD3 T cells showed that all lymphodepleting regimens led to a transient depletion of circulating T cells, with a dose-dependent and more pronounced effect observed within the CPA groups (Fig 6B). Consistently, we detected a proportional and higher engraftment of WT donor CD4 T cells in PB (especially at 2 weeks, Fig 6C and D), spleen, and lymph nodes (Appendix Fig S1A) of all conditioned recipients as compared to non-conditioned mice. After challenge with TNP-KLH, all transplanted mice showed partial rescue of antigen-specific IgG response, at levels comparable to those previously observed in mice infused with uncultured total T cells (Fig 6E). While the highest IgG responses were measured in the high dose CPA group, some mice transplanted after anti-CD4 antibody or ALS treatment also showed high antigen-specific IgG response despite their relatively lower CD4 T-cell engraftment. Nevertheless, only mice treated with high dose of CPA maintained stable amounts of antigen-specific antibodies at long-term follow-up after immunization (Fig 6E).

Overall, these data indicate that transplant of *ex vivo* activated CD4 T cells allows effective rescue of the humoral response in HIGM1 mice and that non-genotoxic conditioning regimens fail to establish robust and long-term responses.

### Primed T cells allow more effective immune responses against pre-experienced antigens

Since a good fraction of T cells in human blood are antigen-experienced memory cells and HIGM1 patients with ongoing infections have circulating T cells already primed by pathogen-specific antigens, we wondered whether adoptive therapy with gene corrected autologous T cells might benefit from the harvest of antigen pre-experienced T cells. To better model this condition, we immunized WT mice with TNP-KLH, collected their CD4 T cells, activated, and grew them *in vitro* and transplanted them in conditioned or not

$Cd40lg^{-/-}$ mice as described before (Fig 6A). Despite both transient leukopenia in recipient mice after CPA treatment (Fig 6F) and engraftment of the infused T cells (Fig 6G and Appendix Fig S1B) were comparable to those observed in the previous experiments, the serum concentrations of antigen-specific IgGs measured after TNP-KLH vaccination were higher for all conditions and similar in mice treated with high or low dose of CPA and, notably, also in several mice transplanted without conditioning (Fig 6H). Moreover, robust secondary responses were observed after a second boost in nearly all mice, except for two non-conditioned mice that lost donor T cells early after infusion and responded poorly also after the first challenge (Fig 6H). These findings further support the therapeutic potential of adoptive T-cell transfer, suggesting that rescue of humoral response against pre-experienced antigens could be favored and possibly achievable even without conditioning regimen.

### CD40LG editing efficiencies achieved in HSPC might provide similar rescue of the humoral immune response in HIGM1 as adoptive T-cell transfer

Having established above (Fig 4) the extent of gene correction achievable in human repopulating HSPC, we wondered whether it was sufficient to rescue the immune function in the $Cd40lg^{-/-}$ mouse model. Since current editing protocols are considerably less efficient and more detrimental for mouse HSPC than for the human counterpart, we used WT mouse HSPC as surrogate of gene corrected cells and transplanted them into $Cd40lg^{-/-}$ recipients together with increasing proportions of $Cd40lg^{-/-}$ HSPC to model the readout of different editing efficiencies (Fig 7A). Chimerism of WT and Cd40lg$^{-/-}$ cells observed within CD11b myeloid cells in PB of transplanted mice over time (Fig EV5A) and within CD19 B cells, CD4 T cells, CD8 T cells, and CD11b myeloid cells of PB and spleen at the end of the experiment closely mirrored that of the input dose of cells, with a mild selective advantage of WT over defective CD4 T cells in the spleen (Fig EV5B). We then vaccinated transplanted mice with Ovalbumin (OVA) and TNP-KLH and evaluated their ability to produce IgG-switched antibodies after the first challenge and a following boost. Mice injected with increasing proportions of WT HSPC displayed a dose-dependent WT CD4 T-cell engraftment (Fig 7B) and rescue of immune function, measured both as serum levels of antigen-specific IgGs and percentages of splenic GC B cells. Whereas mice transplanted with only 1% WT cells nearly failed to produce OVA or TNP-KLH-specific IgGs and to engage B cells for GC formation, mice transplanted with 10% or 25% of WT HSPC partially rescued the switched-antibody responses (Figs 7C and EV5C) and GC formation (Figs 7D and EV5D) to increasing extent. Notably, transplantation of 10–25% WT HSPC led to levels of donor WT CD4 T-cell engraftment and TNP-KLH-specific IgG response comparable to those obtained in $Cd40lg^{-/-}$ mice transplanted either with WT T cells after CPA treatment or with primed WT lymphocytes even in the absence of conditioning (compare Fig 7B with Fig 6C and G; Fig 7C with Fig 6E and H, respectively).

### Comparable protection from a clinically relevant opportunistic pathogen by primed T-cell transfer and HSPC therapy

HIGM1 patients are highly susceptible to infections caused by opportunistic pathogens such as *Pneumocystis jirovecii* (Levy *et al*, 1997;

Winkelstein *et al*, 2003), whose effective clearance requires CD4 T cell-dependent activation of both B cells and macrophages (Otieno-Odhiambo *et al*, 2019). We thus investigated whether transplantation of increasing proportions of WT HSPC or adoptive transfer of WT CD4 T cells into *Cd40lg⁻/⁻* mice would confer protection against the mouse model of Pneumocystis pneumonia (Bishop *et al*, 2012). We intranasally inoculated lung homogenates from *Pneumocystis murina* infected mice into the different groups of transplanted mice, collected their lungs when clinical signs of disease started to appear in the following months and assessed the establishment and progression of infection by quantifying *P. murina* rRNA and lung histopathology.

Typical features of interstitial pneumonia were found in four out of eight mice transplanted with 0% WT HSPC, in one out of seven

mice transplanted with 10% and in two out of seven mice transplanted with 25% functional cells (Appendix Table S2). In contrast, lung sections from mice transplanted with 100% WT HSPC showed very limited evidence of interstitial pneumonia (Appendix Table S2). Consistently, mice with severe interstitial pneumonia showed high burden of *P. murina* rRNA in lung homogenate (Fig 7E), increased lung weight (Fig 7F), and numerous immunostained *P. murina* organisms in alveoli (Fig 7G and Appendix Table S2). Lungs of other mice of these groups showed limited or no presence of the pathogens (Appendix Table S2) and low to undetectable levels of *P. murina* rRNA (Fig 7E). As expected, *P. murina* organisms were absent in mice transplanted with 100% WT HSPC, except for one mouse, which presented very few

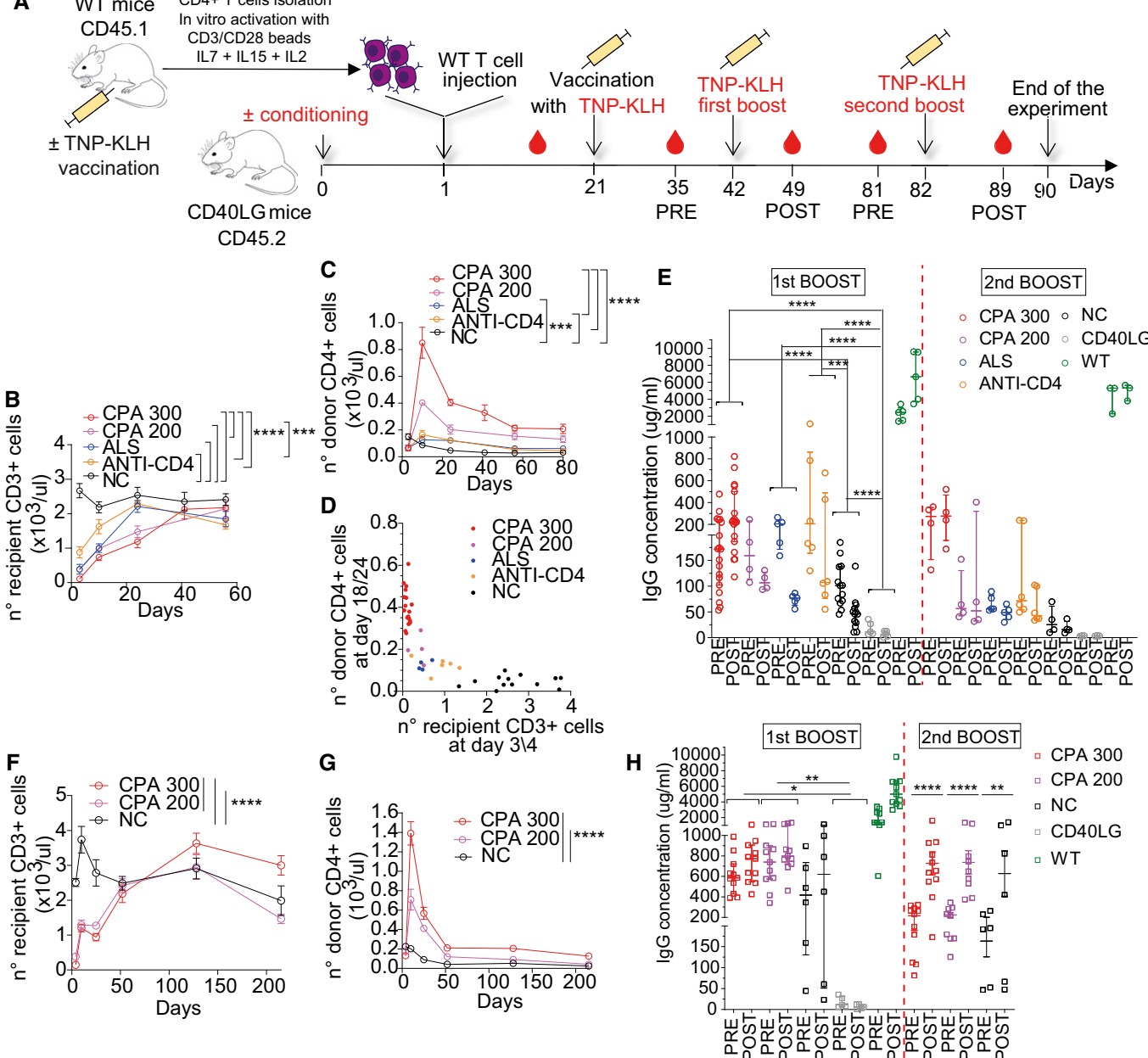

**Figure 6.**

**Figure 6. Rescue of humoral immune response by adoptive transfer of *ex vivo* cultured and *in vivo* primed donor T cells.**

A  Schematics of adoptive transfer of *in vitro* activated WT CD45.1$^+$CD4$^+$ T cells into CD45.2 *Cd40lg*$^{-/-}$ mice pre-conditioned or not with different lymphodepleting regimens and immune response studies.

B  Total counts of recipient CD3$^+$CD45.2$^+$ cells in PB of mice NC (n = 13) or pre-conditioned with 300 mg/kg CPA (n = 19), 200 mg/kg CPA (n = 6), anti-lymphocyte serum (ALS; n = 5), or anti-CD4 antibody (n = 6). Three independent experiments. Comparisons at day 3/4 were performed with an LME model, accounting for multiple experiments, followed by appropriate *post hoc* analysis (see Appendix Supplementary Statistical Methods; ***P = 0.0009 and ****P < 0.0001 in all comparisons). Mean ± SEM.

C  Total counts of engrafted donor CD45.1$^+$ cells in PB of mice from (B). Three independent experiment. Comparisons at day 18/24 were performed with an LME model, accounting for multiple experiments, followed by appropriate *post hoc* analysis (see Appendix Supplementary Statistical Methods; ***P = 0.0002 and ****P < 0.0001 in all comparisons). CPA200 was not included in the analysis because n = 4. Mean ± SEM.

D  Analysis of the relationship between total counts of recipient CD3$^+$ cells gated within CD45.2$^+$ cells at day 3, 4, and engrafted donor CD45.1$^+$ T cells at 18/24 days. This relationship was modeled with an asymptotic NLME model (Appendix Supplementary Statistical Methods), accounting for the different groups and multiple experiments. From the estimated model, a lower value of recipient CD3$^+$ T cells corresponds to a higher value of engrafted donor CD4$^+$ T cells (P = 0.0006).

E  TNP-KLH-specific IgG concentration in sera of mice from (B) collected before (day 35; pre) and after (day 49; post) the first boost (n = 13 NC, 19 CPA 300, 4 CPA 200, 5 ALS, 6 ANTI-CD4; three independent experiments) and at day 81 (pre) and 89 (post) for the second boost (n = 4 NC, 4 CPA 300, 4 CPA 200, 5 ALS, 6 ANTI-CD4). Sera from vaccinated WT (n = 5) and *Cd40lg*$^{-/-}$ (n = 5) mice were used as positive and negative controls, respectively. For early challenge data, comparisons were performed with an LME model, accounting for multiple experiments, followed by appropriate *post hoc* analysis (see Appendix Supplementary Statistical Methods). The reported statistical comparisons refer only to the overall difference among groups (***P = 0.0002 and ****P < 0.0001 in all comparisons). CPA200 was not included in the analysis because n = 4. WT group, not indicated in the figure, is significantly different from all groups (P < 0.0001 in all comparisons). Of note, regarding the comparisons between time-points, the groups CPA 300 and WT show a significant increase between pre- and post-values (P < 0.0001 for both), while all other groups show a significant decrease between pre- and post-values (P < 0.0001 for all except P = 0.0022 for ANTI-CD4). Median ± IQR.

F  Total counts of recipient CD3$^+$ cells gated within CD45.2$^+$ cells after transfer of primed donor T cells in PB of mice treated with 300 mg/kg CPA (n = 11), 200 mg/kg CPA (n = 11) or NC (n = 6). Longitudinal comparisons were performed by LME model followed by an appropriate *post hoc* analysis (Appendix Supplementary Statistical Methods). The reported statistical comparisons refer only to day 4 (****P < 0.0001 in all comparisons). Mean ± SEM.

G  Total counts of engrafted donor CD45.1$^+$ cells in PB of mice from (F). Longitudinal comparisons were performed with an LME model followed by an appropriate *post hoc* analysis (Appendix Supplementary Statistical Methods). The reported statistical comparisons refer only to day 25. At day 215, a significant difference was observed between CPA 300 vs. NC and CPA 300 vs. CPA 200 (****P < 0.0001 for both). Mean ± SEM.

H  TNP-KLH-specific IgG concentration in sera of mice from (F) collected before (day 36, pre) and after (day 50, post) the first boost (n = 6 NC, 11 CPA 300, 11 CPA 200) and at day 212 (pre) and 219 (post) for the second boost (n = 6 NC, 11 CPA 300, 9 CPA 200). Sera from vaccinated WT (n = 11) and *Cd40lg*$^{-/-}$ (n = 5) mice were used as positive and negative controls, respectively. Comparisons were performed with an LME model followed by an appropriate *post hoc* analysis, separately for each challenge data (Appendix Supplementary Statistical Methods). Reported statistical comparisons refer to the overall difference among groups, for early challenge data (*P = 0.0282 and **P = 0.0037), while to differences between time points within each group, for late challenge data (**P = 0.0018 and ****P < 0.0001 in both comparisons). Of note, in the late challenge data, no significant overall differences were observed among the groups. Median ± IQR.

Source data are available online for this figure.

immunofluorescent organisms in the lungs, and they showed absent or almost undetectable levels of *P. murina* RNA (Fig 7E, Appendix Table S2). Overall, while transplanting 100% WT HSPC fully protected mice from infection by *P. murina*, this protection was limited to a fraction of mice when the amount of WT HSPC in the cell product was only in the range of 10–25%.

Concerning mice adoptively transferred with T cells, we harvested the donor cells from WT mice previously co-housed with *P. murina* infected *Cd40lg*$^{-/-}$ mice, thus modeling an autologous source of antigen primed cells. Six of eight control untransplanted *Cd40lg*$^{-/-}$ mice showed high burden of *P. murina* rRNA and increased lung weight (Fig 7H and I), with four of them also showing a typical interstitial pneumonia and numerous immunostained *P. murina* organisms (Fig 7G and Appendix Table S3). On the contrary, the adoptively transferred mice showed very limited evidence of infection. *P. murina* cysts were absent in all adoptively transferred mice (Appendix Table S3), and consistently with these data, these mice also showed absent or almost undetectable levels of *P. murina* RNA as well as normal lung weight (Fig 7H and I).

Overall, these results indicate that both HSPC transplantation and adoptive transfer of primed T cells protect *Cd40lg*$^{-/-}$ mice from a disease-relevant pathogen likely through engaging both humoral and macrophage-mediated responses. However, the T-cell approach was equal if not more effective than HSPC transplantation when the fraction of functional HSPC was limited to represent the levels attainable by current state-of-the-art editing protocol, thus

positioning the former as an attractive new strategy to provide therapeutic benefits in HIGM1.

# Discussion

Here, we developed a gene editing strategy for correcting HIGM1 causing mutations of the *CD40LG* gene in T cells or HSPC with high efficiency and predicted safety and provided a comprehensive set of pre-clinical studies performed in the mouse disease model and with patient-derived cells, which together support the rationale for moving toward clinical testing.

Two previous works showed the possibility to restore function and regulation of *CD40LG* in human CD4 T cells by inserting its cDNA into the 5′ untranslated region of the endogenous gene (Hubbard *et al*, 2016; Kuo *et al*, 2018). However, this design requires incorporation of the full open reading frame and part of the promoter in the corrective donor template; thus, off-target insertion of this sequence might lead to unregulated CD40L expression. To abrogate this risk, we used a 5′-truncated cDNA as editing template and made its expression conditional to targeted insertion. Moreover, by selecting the first intron as nuclease target site, both HDR-mediated and *in sense* NHEJ-mediated integration of the donor template in the target locus can reconstitute a functional gene. The high specificity of the selected gRNA coupled to the use of a high-fidelity Cas9 variant further contribute to minimize the genotoxic risk of the editing procedure. By exploiting state-of-the-art gene editing

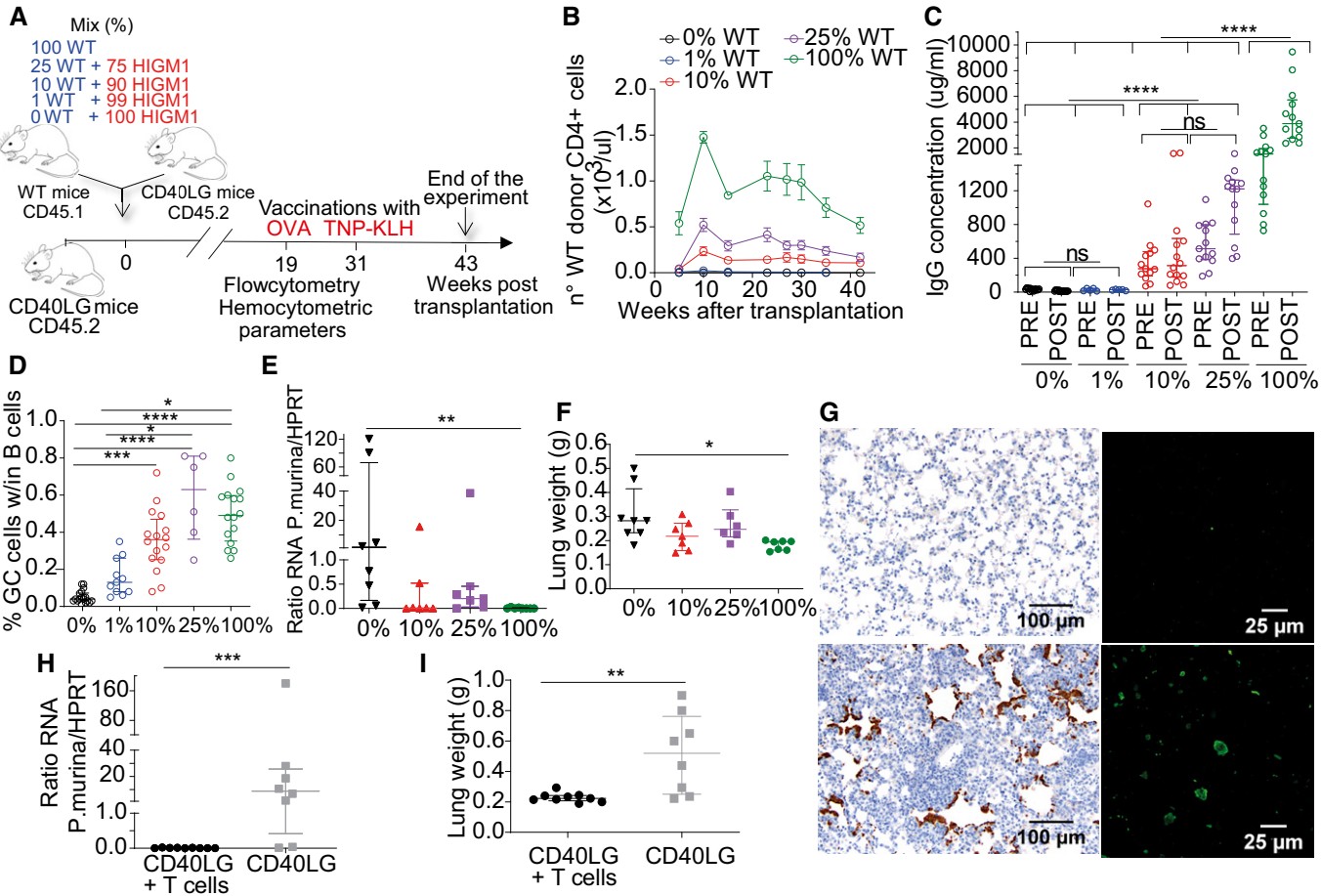

**Figure 7.** *CD40LG* **editing efficiencies achieved in HSPC provide comparable immune rescue as adoptive T-cell transfer.**

A   Schematics of competitive transplant at different ratios of WT (blue) and *Cd40lg*−/− HSPC (red) (100% WT, 25% WT, 10% WT, 1% WT, 0% WT) into lethally irradiated *Cd40lg*−/− recipients.

B   Total counts of WT donor CD4+ T cells in mice from (A) (*n* = 13 100% WT, 8 25% WT, 14 10% WT, 6 1%, 14 0% WT). Two representative experiments shown out of 3. Mean ± SEM.

C   TNP-KLH-specific IgG concentration in sera of mice from (A) collected 7 days before (pre) and after (post) TNP-KLH vaccination (*n* = 13 100% WT, 13 25% WT, 14 10% WT, 6 1% WT, 14 0% WT). Three independent experiments. Comparisons were performed with an LME model, accounting for multiple experiments, followed by an appropriate *post hoc* analysis (Appendix Supplementary Statistical Methods). The reported statistical comparisons refer only to the overall difference among groups (****$P < 0.0001$ in all comparisons). Median ± IQR.

D   Percentage of PNA+GL7+ GC B cells within the spleen of mice from (A) (*n* = 16 100% WT, 6 25% WT, 16 10% WT, 11 1% WT, 18 0% WT). Kruskal–Wallis test followed by *post hoc* analysis with Dunn's test. *P*-values were adjusted with Bonferroni's correction to account for multiple comparisons (*$P = 0.0385$ for 25% WT vs. 1% WT, *$P = 0.0129$ for 100% WT vs. 1% WT, ***$P = 0.0001$ and ****$P < 0.0001$ in both comparisons). Median ± IQR.

E   Quantitation of *P. murina* rRNA in lung homogenate of mice from (A) transplanted with different ratios of WT HSPC cells (*n* = 8 100% WT, 7 25% WT, 7 10% WT, 8 0% WT) and infected with the pathogen. Results are expressed in *P. murina*/HPRT RNA copies. Kruskal–Wallis test followed by *post hoc* analysis with Dunn's test. *P*-values were adjusted with Bonferroni's correction to account for multiple comparisons. Only the groups 100% WT and 0% WT resulted to be significantly different (**$P = 0.0028$). Median ± IQR.

F   Lung weight of mice from (E). Kruskal–Wallis test followed by *post hoc* analysis with Dunn's test. *P*-values were adjusted with Bonferroni's correction to account for multiple comparisons. Only the groups 100% WT and 0% WT resulted to be significantly different (*$P = 0.0254$). Median ± IQR.

G   Left: Representative pictures of lung tissue sections stained immunohistochemically with a rabbit primary antibody (upper left: negative sample; lower left: positive sample). Brown areas represent intra-alveolar aggregates of *P. murina* organisms. Right: Representative pictures of *P. murina* organisms detected by immunofluorescence in lungs homogenate (upper right: negative sample; lower right: positive sample).

H   Quantitation of *P. murina* rRNA in lung homogenate of mice adoptively transferred (*n* = 9 CD40LG+ T cells) or not (*n* = 8 CD40LG) with *in vivo* primed CD4+ T cells in absence of conditioning and infected with the pathogen. Results are expressed in *P. murina*/HPRT RNA copies. Mann–Whitney test. ***P*-value of the comparison = 0.0003. Median ± IQR.

I   Lung weight of experimental mice from (H). Mann–Whitney test. **P*-value of the comparison = 0.0055. Median ± IQR.

Source data are available online for this figure.

protocols, we showed that *CD40LG* correction can be obtained to high and predicted clinically relevant efficiencies in both human T Stem Memory Cells (35%) and long-term repopulating HSPC (25%), while preserving their engrafting capacity. Another improvement of our gene correction design is the inclusion of a selector cassette in the editing template, which in the T-cell approach allows enriching for the edited cells before transplant and possibly depleting them in case of adverse events. Transplantation of a T-cell product enriched in corrected cells will reduce competition in repopulating lymphoid niches and conceivably improve therapeutic benefits. Moreover, since only one *CD40LG* allele is present in male patient cells, this "purifying" selection allows eliminating cells that carry non-HDR rearrangements, such as NHEJ-mediated mutations, deletions, and translocations, or on site integrations of AAV6 fragments (Hanlon *et al*, 2019; Nelson *et al*, 2019). The use of our optimized hEGFRt marker allows coupling selection with the possibility to deplete the transplanted cell product by treatment with a clinically approved monoclonal antibody which, based on the broad clinical experience in tumor therapies, is associated with only minor side effects, such as skin rash (Peréz-Soler & Saltz, 2005; Hansel *et al*, 2010). While our investigation on human cells remains limited in providing direct evidences of T-cell killing, due to the lack of effector cells on xenogeneic models for assessing antibody-dependent cellular cytotoxicity (Shultz *et al*, 1995; Verma *et al*, 2017), previous studies performed in full mouse settings have already proved effective depletion from both blood and solid organs of T cell expressing hEGFRt within 4 days after Cetuximab administration (Wang *et al*, 2011; Paszkiewicz *et al*, 2016). Indeed, this strategy is already under investigation in several clinical trials as safety control of T cell-mediated cancer immunotherapy (Yu *et al*, 2019). Nevertheless, since the depletion by Cetuximab remains a relatively slow process, further studies will be necessary to assess whether this approach would also be suitable for controlling more acute adverse events related to T-cell administration, such as the cytokine release syndrome reported in some patients after the infusion of activated CD8 T cells.

Interestingly, we found that the inclusion in the corrective template of a second cistron for the selection gene stabilized the edited gene mRNA, possibly by enhancing ribosomal recruitment and recycling, through the IRES and the mRNA loop formation (Chou, 2003), respectively, which overall increase translation rate and ribosomal occupancy time on the transcript (Edri & Tuller, 2014). This effect compensated for the lower peak of CD40L expression observed for the edited allele upon T-cell activation, by fully replenishing its intracellular stores without disrupting physiological regulation and extent of surface translocation making it indistinguishable from that occurring from an unedited allele. We investigated the underlying mechanism(s) for the lowered expression of the edited *CD40*LG and found a shorter half-life of the edited vs. unedited mRNA. This outcome might be consequent to transcribing a cDNA rather than a multi-intronic transcript downstream the editing site, although the original intronic sequences remain in the locus and may thus continue to exert cis-acting functions, if present. Since introns and the splicing process *per se* can enhance transcription rate, nuclear export, transcript stability and translation in eukaryotes (Shaul, 2017), we tested different splice acceptor sequences and combination of intervening introns, always with the constrains to fit the maximal repair template size and minimize homology downstream the intended recombination site. However, we did not observe changes in edited gene expression. It remains possible that some unique intron–exon combination of the gene might influence its transcription and/or mark the transcript and impact its translation (Shaul, 2017). We also cannot rule out that gene editing may leave some stable epigenetic marks in the target locus, which affect its transcription and/or translation. Further studies will address this interesting finding, although a lower expression of the edited *CD40LG* was also observed after previous editing attempts, and did not detectably impact the function of edited CD4 T cells in this and previous studies (Hubbard *et al*, 2016), even when we increased the stringency of analysis (see Fig 1K and L). Nevertheless, we cannot rule out that more subtle effects of this lower expression could only emerge on more physiological *in vivo* contexts, which however are difficult to be modeled in pre-clinical studies.

CD40L has a critical function on all CD4 T-cell subsets, but it is also expressed on several other hematopoietic cell types, such as CD8 memory T cells, activated B cells, NK, basophils, and platelets (Van Kooten & Bancherau, 2000). Therefore, we compared in the HIGM1 mouse model the efficacy of cell therapy based either on T cells or HSPC. Since gene editing of mouse hematopoietic stem cells (HSC) and T cells requires different reagents and procedures and achieves lower efficiency than reported for human cells (Schiroli *et al*, 2017), here we used wild-type mouse T cells or HSPC as surrogate models of functional edited cells. Nevertheless, whether edited human cells fully recapitulate upon transplantation the function and long-term persistence of healthy donor cells will have to be determined in clinical studies.

Both T-cell and HSC therapies allowed long-term rescue of the secondary humoral response through induction of memory T and likely B cells, and controlling an opportunistic lung infection, which conceivably requires T cell-dependent activation both of B cells and phagocytes such as alveolar macrophages (Limper *et al*, 1997; Otieno-Odhiambo *et al*, 2019).

Concerning T-cell therapy, prior lymphodepletion increased engraftment of transplanted cells, and this correlated with increased response to vaccination, suggesting competition for homing to T-cell areas of lymphoid organs. Injection of *ex vivo* stimulated CD4 T cells appeared no less effective than that of uncultured cells maybe because stimulation generates memory cells that are endowed with faster activation kinetics in response to antigen compared with naive cells. Accordingly, T cells from vaccinated donors showed significantly higher IgG response upon antigen re-challenge in the recipients than non-specifically activated T cells engrafted at similar level, likely because of enrichment of antigen-experienced cells in the harvest. The increased proportion of antigen-specific memory cells might also explain the lower requirement for preconditioning to rescue humoral responses even at long term and the effective protection of HIGM1 mice from opportunistic infection by the priming pathogen. These findings suggest that T cells harvested from HIGM1 patients with ongoing infections would be enriched for the relevant pathogen specificity and provide immediate and conceivably durable benefits upon correction and reinfusion. Furthermore, these findings might support the rationale of vaccinating patients before T-cell harvest to allow for long lasting antigen-specific immunity upon product infusion. However, since the absence of CD40L could directly affect proper T-cell priming (Grewal *et al*, 1995; Whitmire *et al*, 1999) from antigen presenting cells (Guermonprez *et al*, 2002), further investigation might be required to assess the actual

fraction of antigen-experienced T cell in HIGM1 patients after vaccination.

HSPC therapy might in principle provide broader and more sustained immune reconstitution compared with T-cell transfer and extend correction to other hematopoietic cell types. However, we found that *CD40LG* correction efficiencies achievable on human long-term repopulating HSPC, even if higher than those previously reported for this site (Kuo *et al*, 2018), allowed reaching engraftment levels of functional CD4 T-cells and humoral response reconstitution similar to those obtained by primed T-cell transfer, and provided comparable if not lower protection from a pathogen challenge. Moreover, HSC engraftment requires myelosuppressive conditioning, and while the autologous cell source might allow using reduced intensity regimens, partial chimerism of engrafted cells would further reduce the corrected cell fraction. Moreover, even reduced intensity conditioning may pose significant risks when administered to patients who are adult or in need of lifesaving treatments because affected by uncontrolled infections. On the contrary, T-cell therapy, with or even without lymphodepleting conditioning, would be a better tolerated option in most patients and conceivably the only one available to those with organ damage and uneradicated infections. The T-cell dose used for mouse modeling represents the upper range of cells/kg used in a previous trial with gene-edited CD4 T cells (Tebas *et al*, 2014) and might exceed the doses of pathogen-specific adoptive T-cell therapies used in HSCT settings (Icheva *et al*, 2013).

While care must be taken when translating results from experimental mouse models to the clinical setting, we should acknowledge that humans, as all wild animals, are exposed to commensal and pathogenic microbes throughout their lives, and this microbiome has a profound impact on immune system development, competence, and overall health. The use of laboratory mice housed under specific pathogen-free (SPF) conditions is important to improve experimental consistency, but leaves the mice with an underdeveloped immune system (Huggins *et al*, 2019), thus possibly underestimating the level of immune response to an antigenic challenge predicted for the human setting. Indeed, previous sporadic reports of patients with genetic mosaicism, either an allogeneic HSCT patient with low engraftment (Petrovic *et al*, 2009) or female carriers with skewed X inactivation in the blood (Hollenbaugh *et al*, 1994), would support our contention that even low percentages of CD40L-proficient cells, achieved by either T-cell or HSPC therapy, are sufficient to provide substantial immune protection. A dose escalation design of a T-cell therapy trial will allow safe testing of these predictions in the clinical setting.

The ease of collection, their capacity to expand to large numbers with well-established clinical grade procedures and their long-track of safety upon genetic manipulation make T cells well suitable for a first-in-human testing and validation of a novel gene correction strategy based on engineered endonucleases. Long-term follow-up of clinical studies with genetically engineered T cells have demonstrated persistence of transduced T cells in memory and effector compartments for up to 14 years (Oliveira *et al*, 2015), and similar results have been reported in the more recent trials based on NHEJ-edited T cells (Tebas *et al*, 2014; Stadtmauer *et al*, 2020). Yet, no specific studies have been carried out so far with HDR-edited T cells; thus, the impact of gene editing on the fitness of T cells and, consequently, their long-term persistence still need to be fully ascertained.

Nevertheless, despite functional immune reconstitution may decrease over time and its repertoire may remain constrained by that of the harvest, repeated administrations of the same or a new edited cell product may be performed to prolong therapeutic benefits or broaden specificity. For some HIGM1 patient cohorts, such as pediatric patients potentially eligible to HSCT, but struggling with chronic, poorly controlled infections, a T cell-based editing strategy could be adopted as lifesaving "bridge therapy" to a safer HSCT or HSC gene therapy.

A potential concern for T-cell therapy in HIGM1 is the possibility to elicit an acute inflammatory reaction after rapid reconstitution with functional CD4 T cells in patients with pre-existing infections. Several cases of Immune Reconstitution Inflammatory Syndrome (IRIS) have been reported in the clinical practice when CD4 T-cell immunity has been re-established in a context of immunodeficiency and concurrent opportunistic infections (Barber *et al*, 2012). Dysregulation of the expanding CD4 T-cell subpopulation for a specific pathogen and hyper-responsiveness of the innate immune system to T-cell help were proposed as main causes of pathogenesis (Barber *et al*, 2012). This potential risk could be addressed in clinical trials of T-cell therapy by a dose escalation design or the possibility to rapidly deplete the administered effectors.

Overall, our results provide the scientific rationale, comprehensive pre-clinical safety, and efficacy data and guiding principles toward clinical translation of targeted gene editing for treating HIGM1, and position T-cell therapy as preferred option for first-in-human testing.

## Materials and Methods

### Human samples

Informed consent was obtained from all human subjects. In particular, buffy coats were obtained as anonymized residues of blood donations, used upon signature of specific institutional informed consent for blood product donation by healthy blood donors. Informed consent for biological samples' collection and anonymized biological sample/data sharing for HIGM1 patients was obtained by their own referring physicians, according to local research protocols, reviewed and approved by local ethics committees or institutional review board (IRB). The experiments conformed to the principles set out in the WMA Declaration of Helsinki and the Department of Health and Human Services Belmont Report.

### Primary cells

Peripheral blood mononuclear cells (PBMCs) were freshly purified from Buffy Coat using Ficoll by sequential centrifugations. CD3 T cells were isolated and stimulated using magnetic beads (ratio cell: bead 1:3) conjugated with anti-CD3/anti-CD28 antibodies (Dynabeads human T-activator CD3/CD28, Thermo Fisher). CD4 T cells were isolated by immune-magnetic separation using CD4 T-cell isolation kit (Miltenyi Biotech) according the manufacturer's instructions, and stimulated using Dynabeads. Cells were maintained in Iscove's Modified Dulbecco's Medium (IMDM; Corning) supplemented with 10% FBS (Euroclone), penicillin (100 IU/ml), streptomycin (100 µg/ml), 2% glutamine, and IL-7 (5 ng/ml;

PreproTech) and IL-15 (5 ng/ml; PreproTech). Dynabeads were removed after 6 days of culture. In all the experiments, T cells were derived from male healthy donors.

As regards HIGM1 patients' samples, CD4 T cells were selected after thawing the cryopreserved PBMC according to the protocol described above.

T cells were expanded for 21 days to perform flow cytometry, molecular and functional analyses, and PMA/Ionomycin stimulation. When indicated, NGFR+ edited cells were enriched with MACSelect LNGFR System (Miltenyi Biotech) according to the manufacturer's instructions. Magnetic separation was performed with LD Column.

Human male Cord-Blood (CB) CD34+ cells were obtained frozen from Lonza. Cells were cultured at the concentration of $5 \times 10^5$ cells/ml in serum-free StemSpan medium (StemCell Technologies) supplemented with penicillin (100 IU/ml), streptomycin (100 μg/ml), 2% glutamine, human early-acting cytokines (SCF 100 ng/ml, Flt3-L 100 ng/ml, TPO 20 ng/ml, and IL-6 20 ng/ml; PeproTech), 1 μM SR-1 (Biovision), 50 nM UM171 (Stemcell Technologies), and 10 μM dmPGE2 (Cayman) added only right-after thawing. CB CD34+ cells were expanded for 7 days to perform flow cytometry and molecular analyses. All cells were cultured in a 5% $CO_2$ humidified atmosphere at 37°C.

### Flow cytometry analysis

Cytofluorimetric analyses were performed on FACS Canto II (BD Pharmingen), equipped with DIVA Software and analyzed either with the FSC express software (v. 6, 7, De Novo Software) or Flow Jo Software (FLOWJO, LLC). For surface sample staining, antibodies listed in Appendix Table S4 were used. Single stained and Fluorescence Minus One (FMO) stained cells were used as controls. LIVE/DEAD Fixable Dead Cell Stain Kit (Thermo Fisher), 7-aminoactinomycin (Sigma-Aldrich) was included in the sample preparation for flow cytometry according to the manufacturer's instructions to exclude dead cells from the analysis.

For intracellular staining, surface antigens were stained prior to fixation and permeabilization steps, performed using the BD Cytofix/Cytoperm fixation/permeabilization Kit, according to the manufacturer's instructions.

Cell sortings for specific analysis of edited cells were performed using MoFlo XDP Cell Sorter (Beckman Coulter) or FACS Aria Fusion (BD Biosciences).

### Donor templates and nucleases

AAV6 donor templates for HDR were generated from a construct containing AAV2 inverted terminal repeats, produced by TIGEM Vector Core by triple-transfection method and purified by ultracentrifugation on a cesium chloride gradient. IDLV donor templates for HDR were produced exploiting HIV-derived, third-generation self-inactivating transfer constructs. IDLV stocks were prepared and titered as previously described (Lombardo *et al*, 2007). dsDNA donor templates for HDR were obtained by restriction digestions or synthetized by high-fidelity PCR. dsDNA digestions or amplicons were then purified by gel electrophoresis in order to remove plasmid contamination. Primers for linearization are listed in Appendix Table S5. Schematic designs of each donor construct with homologies for the *CD40LG* gene are reported throughout the paper.

Ribonucleoproteins (RNPs) were assembled by incubating at 1:2 molar ratio either S.p.Cas9 protein (Aldevron) or Hi-Fi Cas9 protein (where indicated; Aldevron) with synthetic cr:tracrRNA (Integrated DNA Technologies) for 10′ at 25°C. Electroporation enhancer (Integrated DNA Technologies) was added prior to electroporation according to manufacturer's instructions, only when treating CD34+ cells.

### Gene editing of primary cells

After 3 days of stimulation, $5 \times 10^5/10^6$ primary T cells derived from both healthy donors and patients were washed with PBS and electroporated with 1.25 μM of ribonucleoprotein (RNP), unless otherwise specified (P3 Primary Cell 4D-Nucleofector X Kit, programs DS-130; Lonza). IDLV transduction was performed 24h before electroporation at MOI 100. AAV6 transduction was performed 15′ after electroporation at a dose of $5 \times 10^4$ vg/cell, unless otherwise specified. dsDNA donors were electroporated along with RNP at a dose of 150 ng.

After 3 days of stimulation, CB CD34+ cells were washed with PBS and electroporated with 2.5 μM of RNP (P3 Primary Cell 4D-Nucleofector X Kit, program EO-100; Lonza). 3 μg of GSE56 mRNA (Schiroli *et al*, 2019) was transfected along with RNP when indicated. AAV6 transduction was performed 15' after electroporation procedure at a dose of $10^4$ vg/cell.

### gRNA screening

Screening of the gRNAs was performed by Editas Medicine. Genomic sequences targeted by the gRNAs are indicated in Appendix Table S6. For the screening, different culture conditions from those reported above have been used. Briefly, T cells were stimulated using Dynabeads (ratio cell:bead 1:1) and maintained in X-Vivo 15 medium (Lonza), supplemented with Human AB Serum, IL-7 (5 ng/ml), IL-15 (5 ng/ml), and IL-2 (100 IU/ml; Proleukin, Novartis Pharma). Dynabeads were removed after 2 days of culture. After 3 days of stimulation, cells were edited and expanded in the same medium, but with a lower IL-2 concentration (50 IU/ml).

### Molecular analyses

For molecular analyses, genomic DNA was isolated either with DNeasy Blood & Tissue Kit or QIAamp DNA Micro Kit (QIAGEN) according to the number of cell available.

Qualitative Polymerase chain reaction (PCR) amplifications were performed using either Go Taq Hot Start Polymerase (Promega) or PfuUltra II Fusion (Agilent), according to the manufacturer's instructions.

Nuclease cutting efficiency was measured by mismatch-sensitive endonuclease assay by PCR-based amplification of the targeted locus followed by digestion with Surveyor Mutation Detection Kit (Integrated DNA Technology) according to the manufacturer's instructions. Digested DNA fragments were resolved and quantified by capillary electrophoresis on LabChip GX Touch HT (Perkin Elmer) according to the manufacturer's instructions.

For digital droplet PCR (ddPCR) analysis, 5–50 ng of genomic DNA were analyzed using the QX200 Droplet Digital PCR System (Bio-Rad) according to the manufacturer's instructions. HDR ddPCR

primers and probe were designed on the junction between the donor sequence and the targeted locus and on control sequences used for normalization (human TTC5 gene, PrimePCR ddPCR Copy Number Assay, Bio-Rad). Thermal conditions for annealing and extension were adjusted as following: 55°C for 1′, 72°C for 2′. For gene expression analyses, total RNA was extracted using either RNeasy Micro Kit (QIAGEN) or RNeasy Mini Kit (QIAGEN), according to the number of cells available. DNase treatment was performed according to the manufacturer's instructions using RNase-free DNase Set (QIAGEN). cDNA was synthetized with SuperScript VILO IV cDNA Synthesis Kit (Invitrogen) with EzDNase treatment. 1–5 ng of cDNA was then used for gene expression ddPCR. Primers and probe were designed on the junction between exons, either for wild-type or codon-optimized edited mRNA sequences. Human IPO8 gene was used as normalizer (PrimePCR ddPCR Expression Probe Assay, Bio-Rad). Thermal conditions for annealing and extension were adjusted as following: 55°C for 1′ and 72°C for 1′.

For *P. murina* detection in lung homogenate, total RNA was extracted using the RNeasy Plus Mini Kit (QIAGEN), according to manufacturer's instructions and retro-transcribed to cDNA using SuperScript VILO cDNA Synthesis Kit (Invitrogen). 0.01–1 ng of cDNA was then used for quantification of *P. murina* burden by ddPCR. Primers and probe were designed to specifically detect *P. murina* rRNA. Murine Hprt gene was used as normalizer (PrimePCR ddPCR Expression Probe Assay, Bio-Rad). Thermal conditions for annealing and extension were adjusted as following: 55°C 1′, no extension time. T-cell spectratyping was performed on 20 ng DNA as previously described (Akatsuka *et al*, 1999). Fragments were resolved and quantified by capillary electrophoresis on LabChip GX Touch HT (Perkin Elmer) according to the manufacturer's instructions.

Primers and probes for PCR and ddPCR amplifications are listed in Appendix Table S5.

**Off-target profiling of candidate gRNA**

Off-target profiling of candidate gRNA was performed by Editas Medicine. Digenome-Seq was adapted from (Kim *et al*, 2015). RNP was prepared as described above. Prior to initiating the cutting reaction, RNP was diluted to 2 μM in 1× H150 with 2 mM MgCl$_2$ and then serial diluted 10-fold. Genomic DNA (Promega) was quantified using Invitrogen's Qubit Broad Range Kit following the manufacturer's protocol and diluted to 140 ng/μl in Low TE. Genomic DNA was then mixed volumetrically 1:1 with 2× H150 buffer 4 mM MgCl$_2$ to prepare a 70 ng/μl. In 96 well plate, 5 μl of 70 ng/μl genomic DNA was mixed with 5 μl of RNP dilution or 1× H150 buffer with 2 mM MgCl$_2$ creating a three-log dose response ranging from 1 to 0.001 μM final for each RNP of interest and an untreated control (12 replicates for each condition). After 16 h of incubation at 37°C, the reaction was terminated by addition of 2 μl of Proteinase K (NEB; 800 units/ml) and subsequent incubation at 55°C for 30 min. To remove excess gRNA, 1 μl of RNase I(f) (NEB; 50,000 unit/ml) was added to each well and incubated at 37°C for 30 min. Cut and control genomic DNA replicates were pooled, purified using 1.8× volume of Agencourt Ampure Xp beads following the manufacturer's protocol and 1 μg of genomic DNA from each condition was submitted for PCR-free whole genome 2 × 150 sequencing at Genewiz NGS South Plainfield, NJ.

"GUIDE-Seq primers" are listed in Dataset EV1. GUIDE-Seq (Tsai *et al*, 2015) process was performed with the UDiTaS protocol (Giannoukos *et al*, 2018) with the following modifications. Briefly, two tagmentation reactions were set up for each sample using 50 ng DNA, 4 μl nuclease free water, 2 μl 1 mg/ml transposome (Tn5 enzyme complexed with a custom barcoded double-strand oligo), 4 μl 5× TAPS-DMF buffer (50 mM TAPS-NaOH at pH 8.5, 25 mM MgCl$_2$, 50% dimethylformamide), and 10 μl DNA (5 ng/μl). Reactions were incubated at 55°C for 7 min, placed on ice, and cleaned up with Zymo's 96-well PCR purification kit, ZR-96 DNA Clean & Concentrator-5, according to the manufacturer's protocol with the following exceptions: 170 μl of binding buffer was added to each reaction and 25 μl of nuclease free water was added to each column for elution. The first PCR step was performed as following: 25 μl 2× Platinum SuperFi Master mix (Thermo Fisher Scientific), 3 μl 0.5 M Tetramethylammonium chloride (TMAC; Sigma-Aldrich), 1.25 μl 10 μM P5 primer (OLI5589), 0.375 μl 100 μM gene-specific minus stand primer (OLI6937) or plus strand primer (OLI6938), and 20 μl tagmented DNA, using amplification protocol 98°C × 2′, (98°C × 10 s, 65°C × 10 s, 72°C × 90 s) × 15 cycles, 72°C × 5′. Amplicons were purified with AMPure XP beads (0.9×) according to the manufacturer's protocol and eluted in 15 μl nuclease-free water directly into the second PCR step, performed as following: 25 μl 2× Platinum SuperFi Master mix (Thermo Fisher Scientific), 2.5 μl 10 μM P5 primer, 7.5 μl 10 μM P7_barcode_SBS12 primer, using amplification protocol 98°C × 2′, (98°C × 10 s, 65°C × 10 s, 72°C × 90 s) × 16 cycles, 72°C × 5′. Amplicons were purified with AMPure XP (0.9×) according to the manufacturer's protocol and run on the Agilent Tapestation for quantification and sizing of the products to calculate nM for pooling. Four samples (or eight reactions) were pooled. AMPure XP clean-up was increased to 1.2× reaction volume after pooling and to 1.5× reaction volume after size selection on BluePippin (250–1,000 bp). Library quantification was performed using Qubit™ dsDNA HS Assay Kit (Thermo Fisher Scientific). The sequencing library (9 pM) was loaded into an Illumina MiSeq Reagent kit v2 to obtain 2 × 150 cycle reads. Sequencing reads were demultiplexed, and paired reads were merged using PEAR and adapters removed using cutadapt. Reads were aligned to the appropriate reference genome using Bowtie2 (Langmead & Salzberg, 2012) and samtools (Li *et al*, 2009) were used to create and index sorted bam files. In the final step, sites in the genome with high quality alignments were classified as bidirectional if they contained reads in at least two of the following conditions: (i) sense reaction and alignment in plus strand, (ii) sense reaction and alignment in minus strand, (iii) antisense reaction and alignment in plus strand, and (iv) antisense reaction and alignment in the minus strand (Tsai *et al*, 2015). Finally, needle (Rice *et al*, 2000) was used to perform a global alignment between the site and the gRNA. This step identified the number of mismatches and bulges for each candidate off-target site. If a control sample was also sequenced, the background counts were calculated in all the regions where reads for the gRNA of interest were found. Output from the analysis contains the regions of the genome where reads were found. The final list of candidate off-target sites was comprised of the sites with at most six combined mismatches or bulges, and that have reads classified as bidirectional in at least one experiment.

For the screening of 93 OT, primers (listed in Dataset EV1) were designed to the candidate editing target sites using a custom

wrapper built around the software package Primer3 (Untergasser et al, 2012). The first PCR step incorporates gene-specific primers that include 5′ handles on both the forward and reverse primers, while the second PCR step adds the full-length Illumina sequencing primers and sample barcodes. The first PCR step was performed in a 12 μl reaction volume, consisting of 6 μl of NEBNext® Ultra™ II Q5® Master Mix (New England Biolabs), 0.25 μM forward and 0.25 μM reverse primer, and 20 ng of gDNA template, using amplification protocol 98°C × 30 s, (98°C × 10 s, 60°C × 15 s, 72°C × 30 s) × 20 cycles, 72°C × 5′. The second PCR step was performed in a 12 μl reaction volume, consisting of 6 μl of NEBNext® Ultra™ II Q5® Master Mix (New England Biolabs), 1.0 μM forward and 1.0 μM reverse primers, and 4 μl step 1 product, using amplification protocol 98°C × 30 s, (98°C × 10 s, 60°C × 15 s, 72°C × 30 s) × 14 cycles, 72°C × 5′. PCRs were pooled and purified using (0.9×) Agencourt AMPure XP beads (Beckman Coulter Agencourt AMPure XP—PCR Purification) following manufacturer's protocol and were selected by 300–1,200 bp size on the BluePippin (Sage Science, Beverly, MA). Another purification with (0.9×) Agencourt AMPure XP was performed. 8–10 pM of sequencing library was loaded into a MiSeq Reagent kit v3 to obtain 2 × 300 cycle reads. Analysis of indel rates was done using custom python, R, and shell scripts that called standard NGS sequence analysis software tools. Briefly, fastq files for each sample were generated by demultiplexing the run with the software installed on the Illumina MiSeq sequencers. For each sample, Bowtie2 (Langmead & Salzberg, 2012) was used to build index files from its corresponding amplicon sequence. Afterward, forward and reverse paired-end reads were aligned to the amplicon sequence using Bowtie2 or the python module difflib. Next, all alignment files are analyzed using samtools (Li et al, 2009) and use the cigar string for each read to locate all insertions and deletions. Finally, given a window of interest within the amplicon (± 2 bases on either side of the expected cut site), number of distinct reads are counted with insertions and deletions within the window. Then, the indel fraction are calculated as the number of distinct reads with an indel in the window divided by the number of distinct reads.

## Off-target assessment with WT or HF-Cas9: library preparation and analysis

PCR amplicons for each individual sample were generated starting from 100 ng of purified DNA by nested PCR using primers listed in Appendix Table S5. The first PCR step was performed with PfuUltra II Fusion (Agilent) according to the manufacturer's instruction, using the following amplification protocol: 95°C × 2′, (95°C × 25 s, 55°C × 25 s, 72°C × 20 s) × 25 cycles, 72°C × 5′. Amplicons were separately purified using MinElute PCR Purification kit (QIAGEN). The second PCR step was performed with PfuUltra II Fusion, using the following amplification protocol: 95°C × 2′, (95°C × 25 s, 56°C × 25 s, 72°C × 20 s) × 25 cycles, 72°C × 5′. Second-step PCR primers were endowed with tails containing P5/P7 sequence, i5/i7 Illumina tag to allow multiplexed sequencing, and R1/R2 primer complementary sequence. Amplicons were separately purified with AmpPure XP beads (Invitrogen). Library quality was assessed by Agilent Tape Station (Agilent Technologies). Finally, amplicons were multiplexed and run on MiSeq 2 × 300 bp paired end (Illumina).

Samples were then analyzed with CRISPResso2, a suite of software developed to detect indel events in off-target sequences. In

details, the CRISPRessoBatch pipeline was used to filter NGS reads relying on the phred33 score, getting rid of low quality sequences, and then to remove Illumina TruSeq3-PE adapters using Trimmomatic (http://www.usadellab.org/cms/?page=trimmomatic). After that, each couple of paired-end reads was merged using FLASh to produce a single sequence, which was then mapped to the input amplicon reference sequence (using a global alignment method). A single guide RNA (sgRNA) was also provided to CRISPResso2 to focus the analysis of the variant identifications and quantifications on that region. As reported in the guidelines of CRISPResso2, the sgRNA (usually 20nt) needs to be provided immediately adjacent to but not including the PAM sequence. Finally, the CRISPRessoCompare utility was employed to compare each sample with the corresponding UT control, in order to identify the differences (indel events) between the two conditions in comparison with the reference amplicon. The quantification of the indels in each position of the amplicon was reported by CRISPRessoCompare in terms of percentage of reads supporting that event and corresponding P-value, resulting from the comparison with the corresponding UT sample, which was then used to identify the significant events.

## PMA/Ionomycin and Actinomycin D treatments

From 6 to 12 days after electroporation, T cells were stimulated for 5 h with Phorbol-12-myristate-13-acetate (PMA, 10 ng/ml; Calbiochem) and Ionomycin (500 ng/ml, Sigma-Aldrich) in cytokine-free medium, then washed and cultured in complete medium. Surface expression of CD40L on T cells was followed over a 24/48-h time course by serial flow cytometry analyses (0, 3, 6, 8, 24, 48 h after activation). When indicated, 6 h after PMA/Ionomycin stimulation T cells were treated with Actinomycin D (10 μg/ml, Sigma-Aldrich) to inhibit transcription.

## Ex vivo functional studies of edited T cells

Naive B cells were isolated from peripheral blood mononuclear cells by immune-magnetic negative selection using human naive B-cell isolation kit II (Miltenyi Biotec), according to the manufacturer's instructions.

All cells were cultured in RPMI 1640 (CORNING) containing 1% of penicillin/streptomycin (P/S) (Thermo Fisher scientific), 20% FBS, 20 mM N-2-hydroxyethylpiperazione-N′-2-ethansulfonic acid (HEPES) (both from Sigma-Aldrich), 1% L-glutamine (Life Technologies) and 55 μM 2-mercaptoethanol (Gibco—Life Technologies).

Prior to co-culture, CD4 T cells were washed and rested overnight in cytokine-free medium. T cells were then activated for 5 h using two different stimuli:

1 CD3/CD28 Dynabeads at a 1:1 bead:cell ratio;
2 PMA (1 ng/ml, Sigma-Aldrich) and Ionomycin (500 ng/ml, Sigma-Aldrich).

After removing Dynabeads and PMA/Ionomycin, and washing cells with complete medium, B and T cells were co-cultured in 200 μl of medium previously described at a 1:1 B-cell:T-cell ratio in a 96-well plates flat bottom (CORNING).

T and B cells were kept in culture for 5 days in the presence of IL-2 (50 ng/ml), IL-7 (5 ng/ml), and IL-15 (5 ng/ml) (all from PeproTech) and B cells were stimulated using various combination

of the following cytokines: IL-21 (100 ng/ml) (PeproTech), human Toll-like receptor 9 (TLR9) agonist CpG oligodeoxinucleotide (ODN) 1826 (2.5 μg/ml) (InvivoGen), anti-IgA[+]IgG[+]IgM (15 μg/ml, goat anti-human IgA[+]IgG[+]IgM) (Jackson ImmunoResearch), and soluble CD40L (3 μg/ml) (ENZO Life Sciences).

In order to analyze B-cell class-switch recombination, after 5 days of co-culture cells were recovered, counted, and analyzed by FACS after staining for CD4, CD69, CD19, IgM, and IgG.

To evaluate B-cell proliferation, naive B cells were labeled with CellTrace™ Violet Cell Proliferation Kit (Thermo Fisher Scientific) following the manufacturer's instructions and then co-cultured with T cells following the protocol previously described. The proliferation was analyzed after 5 days of T/B co-culture by FACS.

Immunoglobulin-secreting cells were analyzed 5 days after co-culture by ELISPOT assay performed in plates with nitrocellulose membrane (Merck Millipore) coated with anti-IgG or anti-IgM (both from Southern Biotech). After blocking with PBS (CORNING) and 1% BSA (Sigma-Aldrich), serial dilution of total cells (from $0.5 \times 10^4$ to $0.25 \times 10^4$) were added and incubated ON at 37°C. Plates were then incubated with isotype-specific secondary antibodies (both from Southern Biotech), followed by streptavidin-HRP (Thermo Fisher) and finally developed with 3-amino-9-ethylcarbazole (Sigma-Aldrich) as a chromogenic substrate. Plates were scanned and counted using the Automated ELISA-Spot Assay Video Analysis System (AELVIS) to determine the number of spots/wells.

### In vitro depletion

For *in vitro* depletion, 5,000 T cells/well were seeded in complete medium in 96-well round bottom plate. The αEGFR-SAP immunotoxin was prepared by combining biotinylated Cetuximab antibody (clone #Hu1, R&D Systems) with streptavidin-SAP conjugate (2.3 saporin molecules per streptavidin, Advanced Targeting Systems) in a 1:1 molar ratio and diluted in PBS at two doses (5 nM, 1 nM) immediately before use. Immunotoxin, and the same quantity of antibody alone or toxin alone were added to cells for 3 days and then lymphocytes were collected for flow cytometry analysis.

### Mice

C57Bl/6 Ly45.1 and C57Bl/6 Ly45.2 mice were purchased from Charles River Laboratory, and $Cd40lg^{-/-}$ (B6.129S2-Cd40lgtm1Imx/ J) and NOD-SCID-IL2Rg/ (NSG) mice were purchased from The Jackson Laboratory. C57Bl/6 Ly45.1 or C57Bl/6 Ly45.1/Ly45.2 obtained by crossing C57Bl/6 Ly45.2 and C57Bl/6 Ly45.1 mice at the San Raffaele Scientific Institute animal research facility, were used as donors for adoptive T-cell transfer and HSPC transplant into $Cd40lg^{-/-}$ mice. All the mice were maintained in specific pathogen-free (SPF) conditions, and all animal procedures were designed and performed with the approval of the Animal Care and Use Committee of the San Raffaele Hospital (IACUC #749, #818) and communicated to the Ministry of Health and local authorities according to Italian law.

### CD4 T-cell xenotransplantation studies in NSG mice

From day 12 to day 20 of culture, $10^7$ CD4 T cells (untreated, bulk treated for editing or positive fraction of enriched edited cells) were injected intravenously into 7- to 10-week-old male NSG mice. Sample size was determined by the total counts of available treated cells. Attribution of mice to each experimental group was random. Human CD45[+]CD4[+] cell engraftment and the presence of gene-edited cells were monitored by serial collection of blood from the mouse eye. Mice were weekly monitored and weighed to eventually observe appearance of sign of graft vs. host disease. At the end of the experiment (15 weeks after transplantation), spleen was harvested and analyzed. PMA/Ionomycin stimulation was performed on CD4 T cells purified from the spleen of engrafted NSG mice.

### HSPC xenotransplantation studies in NSG mice

In all the experiments, HSPC from multiple male donors were pooled together to reach sufficient number of cells. We treated the same starting number of HSPC and transplanted the total outgrowth in NSG mice at day 4 of culture. Cells were injected intravenously into 8-week-old female NSG mice, 4 h after sublethal irradiation (150–180 cGy). Sample size was determined by the total counts of available treated cells. Attribution of mice to each experimental group was random. Human CD45[+] cell engraftment and the presence of gene-edited cells were monitored by serial collection of blood from the mouse eye and, at the end of the experiment (20 weeks after transplantation), BM, spleen, and thymus were harvested and analyzed. PMA/Ionomycin stimulation was performed on CD4 T cells purified from the spleen of engrafted NSG mice.

### Murine T-cell transplantation studies

For adoptive T-cell transfer of naive T cells, CD3 lymphocytes were harvested from the spleen of 8-week-old male C57Bl/6 Ly45.1 mice by immune-magnetic separation (Pan T cell isolation kit, Miltenyi Biotec). Right after purification, $2 \times 10^6$, $10^7$ or $2 \times 10^7$ naive T cells were intravenously injected in $Cd40lg^{-/-}$ mice pre-conditioned or not with 300 μg/g of cyclophosphamide (CPA, Baxter) 1 day before T-cell infusion. For adoptive T-cell transfer of activated T cells, CD4 lymphocytes were harvested from the spleen of 8-week-old C57Bl/6 Ly45.1 mice by immune-magnetic separation (CD4 T-cell isolation kit, Miltenyi Biotec) and subsequently activated using magnetic beads (ratio cell:bead 1:1) conjugated with anti-CD3/anti-CD28 antibodies (Dynabeads mouse T-activator CD3/CD28; Thermo Fisher). T cells were cultured in RPMI supplemented with 10% Fetal Bovine Serum (FBS, Euroclone), penicillin (100 IU/ml), streptomycin (100 μg/ml), 1% glutamine, IL-2 (30 U/ml), IL-7 (5 ng/ml), IL-15 (5 ng/ml), Sodium Pyruvate (1 mM Thermo Fisher Scientific), Hepes (20 mM Thermo Fisher Scientific), MEM Non-Essential Amino Acids (Thermo Fisher Scientific 1 μM), and Beta-Mercaptoethanol (0.05 mM Thermo Fisher Scientific). Activated CD4 T cells were expanded in culture for 7 days prior to infusion in mice at a dose of $10^7$ cells. Mice were pre-treated with intraperitoneal (i.p.) injection of 200 or 300 μg/g of CPA. CPA was administered in one single large dose or a preliminary ("priming") dose of CPA at 50 μg/ g was given 7 days before a single large dose of 150 μg/g or 250 μg/g to reduce lethality in mice (Collis *et al*, 1980). The large dose was administered 1 day before T-cell infusion. 50 μg anti-CD4 antibody (BioXcell) or 250 μg ALS (Thymocyte Antibody, LSBio)

were given i.p. 7 days before T-cell infusion. Serial collections of blood from the mouse tail were performed to monitor the hematological parameters and donor T-cell engraftment. At the end of the experiment, spleen and lymph nodes were harvested and analyzed.

### In vivo immunization and IgG quantification

Mice were immunized i.p. with 100 μg of TNP-KLH (Lgc Biosearch Technologies) in Imject Alum Adjuvant (1:2) (Thermo Fisher Scientific) or with 40 μg OVA (Sigma-Aldrich) in Freund's Adjuvant, Incomplete (Sigma-Aldrich). Serum was collected at day 0, 7, 14, and 21 after immunization. Mice were boosted as described above on day 27 or 28, and serum was collected on day 7 after re-challenge. For IgG quantification, the concentration of antigen-specific IgGs in mouse sera was determined by an enzyme-linked immunosorbent assay (ELISA). Plates were coated with 100 μl/well of either 5 μg/ml TNP-KLH or 2 μg/ml OVA in carbonate buffer. Following incubation, plates were washed three times in PBS containing 0.05% Tween20 (Sigma-Aldrich) (Wash Buffer). The plates were then blocked for 1 h using 100 μl/well of PBS containing 1% Bovine Serum Albumine (BSA), followed by a washing step, as described above. Serum samples were serially diluted in wash buffer and 100 μl per well of each diluted sample was added into the plate and incubated for 2 h at room temperature. For determination of the plate background optical density (OD) values, some wells were incubated with wash buffer alone. Following incubation, plates were washed and 100 μl/well of HRP-conjugated goat anti-mouse (Southern Biotech 1:10,000) was added and incubated for 1 h at room temperature. After washing, the plates were incubated for 5′ with 3,3′,5,5′-tetramethyl benzidine (TMB, Sigma-Aldrich) substrate at room temperature. The reaction was stopped by the addition of 50 μl of 1 M $H_2SO_4$. The OD values at 450 nm were determined for each well using a Multiskan GO microplate reader (Thermo Fisher Scientific) and normalized to IgG1 standard curves. Results were expressed as mean of duplicate determinations.

### Germinal center quantification

Tissue samples were formalin fixed and paraffin embedded. After peroxidase block (20 min) and rodent block (30 min; Biocare Medical), paraffin sections (1.5 μm) were incubated in PNA (biotinylated peanut agglutinin anti-mouse B-1075; Vector Laboratories; 1:600) 1 h and in 4plus Streptavidin-HRP Label (Biocare Medical) 20 min with final DAB (diaminobenzidine, Biocare Medical) development and counterstaining nuclei with H&E.

Digital images were acquired with Olympus DP70 camera mounted on Olympus Bx60 microscope, using CellF Imaging software (Soft Imaging System GmbH). Analysis was performed using Olympus Slide Scanner VS120-L100 to acquire digital images and Image-pro software.

### ELISPOT assay with murine splenocytes

The ELISPOT was performed in nitrocellulose membrane 96-well flat-bottomed plate (Millipore) coated with 10 μg/ml of TNP-KLH. After blocking with PBS 1% BSA, serial dilutions of total murine splenocytes (from $2 \times 10^5$ to $2.5 \times 10^3$ cells/well) were incubated overnight at 37°C/5% $CO_2$. Plates were washed and antigen-specific IgG-producing cells were detected by isotype-specific secondary detection monoclonal antibody and streptavidin-HRP, and finally developed with 3-amino-9-ethylcarbazole (Sigma-Aldrich) as chromogenic substrate. Number of spots per well were counted by ImmunoSpotR S6 Fluorescent Analyzer (Cellular Technology Limited).

### Murine HSPC transplantation studies

Donor mice between 6 and 10 weeks of age were euthanized by $CO_2$, and BM cells were retrieved from femurs, tibias, and humeri. HSPC were purified by Lin- selection using the mouse Lineage Cell Depletion Kit (Miltenyi Biotec) according to the manufacturer's instructions. Cells were then cultured in serum-free StemSpan medium (StemCell Technologies) containing penicillin, streptomycin, glutamine, and a combination of mouse cytokines (20 ng/ml IL-3, 100 ng/ml SCF, 100 ng/ml Flt-3L, 50 ng/ml TPO all from PeproTech), at a concentration of $10^6$ cells/ml. For competitive transplants, C57BL/6-Ly5.1 and $Cd40lg^{-/-}$ (CD45.2) Lin- cells were cultured for 16 h in the medium described above, mixed at the indicated ratios, and transplanted at a total dose of $10^6$ cells/mouse into 8-week-old lethally irradiated $Cd40lg^{-/-}$ CD45.2) mice. Serial collections of blood from the mouse tail were performed to monitor the hematological parameters and donor cell engraftment. At the end of the experiment, BM, thymus and spleen, and lymph nodes were harvested and analyzed.

### In vivo protection against Pneumocystis murina

Pneumocystis murina organisms were isolated from lungs of $Cd40lg^{-/-}$ mice previously inoculated with P. murina or by co-housing them with a $Cd40lg^{-/-}$ seeder mouse infected with a large load of P. murina. $Cd40lg^{-/-}$ mice showing symptoms of P. murina induced pneumonia, including weight loss of approximately 30%, hunched-back posture, listlessness, and rapid shallow breathing, were euthanized and murine lungs were collected to quantify infection and obtain the infectious lung homogenate. Briefly, lungs were homogenized in 10 ml homogenization buffer (58.5 mM disodium phosphate, 1.5 mM monopotassium phosphate, 43.5 mM sodium chloride, 10 mM trisodium citrate, 10 mM dithiothreitol and 2.7 mM potassium chloride, pH 7.4) or PBS, in Miltenyi gentleMACS dissociator, then filtered through a 40 μm nylon filter and centrifuged at 2,330 g for 10 min to remove large debris. Pellet was resuspended in 1 ml of RPMI-1640 with 10% FBS and 7.5% DMSO for cryopreservation. For infection, lung homogenate was thawed in RPMI-1640 with 20% FBS and pellet was resuspended in 50 μl/mouse of PBS for intra-nasal administration. Pulmonary lesions suggesting P. murina-induced pneumonia and P. murina organisms themselves were detected on formalin-fixed, paraffin-embedded left lung specimens by hematoxylin and eosin (H&E) staining and immunohistochemistry (IHC). For this latter procedure, 5- to 10-μm lung sections were immunostained with an anti-P. murina primary antibody raised in rabbit (Bishop et al, 2012) and by the avidin-biotin-peroxidase (ABC) procedure (VECTASTAIN® Elite ABC-Peroxidase Kit Standard, Vector Laboratories).

### The paper explained

#### Problem

X-linked hyper-IgM syndrome type I (HIGM1) is a primary immunodeficiency caused by inactivating mutations in the CD40 ligand gene (*CD40LG*) that mainly impair the T cell helper function to B cells and macrophages. This disease represents a suitable candidate for a gene correction strategy because preclinical studies of Hematopoietic Stem Cells (HSC) gene therapy have already shown i) evidence of potential efficacy even with small input of transduced cells; ii) safety issues due to unregulated transgene expression. Recent studies proposed T cell or HSC gene editing as a potentially safer alternative to restore *CD40LG* function while preserving its physiologic regulation. It remained however unclear if T-cell therapy can effectively correct HIGM1 phenotype and if the low gene editing efficiency obtained until now in HSC can be sufficient to rescue the disease.

#### Results

We designed a CRISPR/Cas9-based gene editing strategy aimed to insert a 5′-truncated corrective *CD40LG* cDNA within the first intron of the human endogenous gene, effectively making the expression conditional to targeted insertion in the intended locus, thus improving the expected safety of the editing strategy compared to those previously reported. By exploiting a protocol that preserves long term surviving T stem memory cells, we reproducibly obtained ~35% of editing efficiency in both healthy donor and patients derived T cells, restoring a regulated, although partial, CD40L surface expression. Nevertheless the level of expression obtained in edited CD4 T cells was sufficient to fully restore their helper function to B cells. In order to select, track and eventually deplete edited cells, we coupled the corrective cDNA with a clinically compatible selector gene and, surprisingly, increased also the surface expression of CD40LG to physiological levels, maintaining its regulation. We then broadened application of the gene editing strategy to HSPC, and obtained stable ~30% editing after xenotransplantion in NSG mice by exploiting our recently optimized gene editing protocol. Finally, we evaluated the therapeutic potential of both T cell and HSPC therapies into HIGM1 mice, infusing wild-type murine cells as surrogate models of functional edited cells. Administration of functional T cells at doses representative of those used in adoptive T cell therapy into HIGM1 mice pre-conditioned or not with different lymphodepleting regimens achieved long-term, stable T cell engraftment and partial rescue of antigen-specific IgG response and germinal center formation in splenic follicles after vaccination with a thymus dependent antigen (TNP-KLH). Remarkably, infusion of T cells from mice previously exposed to the antigen, better modeling the harvest of autologous cells from patients, was effective even in the absence of conditioning and protected the mice from a disease-relevant infection induced by the opportunistic pathogen *Pneumocystis murina*. Interestingly, by transplanting a 10-25% proportion of functional HSC along with HIGM1 ones in HIGM1 mice, modelling the editing efficiencies achieved in human HSC with our optimized conditions, we observed a rescue of immune functions comparable to that of T-cell therapy.

#### Impact

Overall, our findings suggest that autologous edited T cells could provide immediate and substantial benefits to HIGM1 patients and position T-cell ahead of HSPC gene therapy because of easier translation, lower safety concerns raised by T cell manipulation and potentially comparable clinical benefits. These results establish the rationale and guiding principles for supporting a first-of-this-kind clinical trial of the proposed therapeutic approach for treating HIGM1.

*Pneumocystis murina* organisms were also detected in lung homogenate by indirect immunofluorescence staining (IF) using an anti-*P. murina* primary antibody raised in rabbit and Alexa Fluor 488–labeled goat anti-rabbit IgG (Thermo Fisher Scientific) as secondary antibody. *P. murina* ribosomal RNA in lung homogenate of infected mice was quantified by ddPCR, as previously described (Molecular Analyses Section).

### Statistical analyses

Correlations between numerical variables were evaluated with Spearman's correlation coefficient. Standard comparisons of numerical variables between two groups were evaluated either with the Mann–Whitney test, in case of independent groups, or with Wilcoxon's paired test, in case of paired groups. Standard comparisons of numerical variables between more than two groups were performed either with Kruskal–Wallis test, in case of independent groups, or with Friedman test, in case of paired groups, followed by a *post hoc* analysis with Dunn's test. When comparisons needed to account for more complex dependencies (e.g., considering also multiple experiments or multiple donors or different number of replicates), appropriate linear mixed-effects (LME) models were estimated, eventually followed by an appropriate *post hoc* analysis with the R package phia, as described in the corresponding Figure Legend or in Appendix Supplementary Statistical Methods. For the comparisons among groups in case of longitudinal data, appropriate mixed-effects models were applied, as detailed described in Appendix Supplementary Statistical Methods. Briefly, when the trend showed a linear or a nonlinear trajectory with a shape following a known parametric function, the appropriate linear or nonlinear mixed-effects (NLME) regression, respectively, was performed with the time treated as a continuous variable. Final models were obtained with a backward variable selection. In the other cases, an LME regression was estimated with the time treated as a categorical variable and *post hoc* comparisons at each time-point were performed with the R package phia. In all the mixed-effects analyses, when necessary, to meet the assumptions of the model, an adequate transformation of the dependent variable was used.

In all *post hoc* analysis or, in general, when the analysis involved multiple comparisons and/or testing, the $P$-values were adjusted with Bonferroni's correction. In all the analyses, $P$-values less than 0.05 were considered significant (*$P < 0.05$, **$P < 0.01$, ***$P < 0.001$, ****$P < 0.0001$. "ns" means non-significance). $P$-values of LME and NLME model analyses are reported in the Appendix Supplementary Statistical Methods. All statistical analyses were performed using R 3.5.0 (http://www.R-project.org/) or GraphPad Prism v8. Inferential techniques were carried out whenever appropriate sample size was available and they were necessary for the interpretation of the data, otherwise descriptive statistics are reported.

## Data availability

Additional data are available online in the Expanded View and Appendix. This study includes no data deposited in external repositories.

**Expanded View** for this article is available online.

## Acknowledgements

We thank L. Notarangelo (NIAID, NIH, Bethesda, Maryland, US), A. Aiuti, D. Canarutto (SR-Tiget), and all members of LN's laboratory for discussion, the

IRCCS San Raffaele Hospital Flow Cytometry facility (FRACTAL), the Center for Omics Sciences, IRCCS San Raffaele Scientific Institute, E. Ayuso (Translational Vector Core UMR1089, Nantes), A. Auricchio and M. Doria (Telethon Institute of Genetics and Medicine; TIGEM, Pozzuoli (NA), Italy) for providing AAV6 vectors, J.A. Kovacs and L.R. Bishop (NIH Clinical Center) for providing *P. murina* organisms and helping in setting up the infection model together with G. Sitia and D.M. Cirillo (IRCCS San Raffaele Scientific Institute), G.M Cavestro (IRCCS San Raffaele Scientific Institute) for help with Cetuximab supplying. We thank S. Ferrari, L. Bertaggia, M.A.J.M. Hendriks, and V. Capo for help with some experiments, T. Plati for technical support in ddPCR analyses, L. Sergi Sergi for the help with IDLV production, A. Annoni for advice on OVA vaccination studies (SR-Tiget), P. Romanelli for the help in performing histopathology of *P. murina*-infected mice (Fondazione Unimi), C. Di Serio for coordinating CUSSB support (Vita-Salute San Raffaele University), S. Tantou ("Aghia Sophia" Children's Hospital), J. Chou and R.S. Geha (Boston Children's Hospital) for helping MK in genotyping one HIGM1 patient, M.E. Bernardo (SR-Tiget and San Raffaele Hospital), A. Finocchi and A. Scarselli (Bambino Gesù Children's Hospital, Rome), S. Volpi (IRCCS Giannina Gaslini), and A. Schulz (University Medical Center Ulm) for making available some patient-derived samples, which it was not possible to use for this work. This work was initially supported by a sponsored research agreement with Editas Medicine terminated in 2018 and by grants to: PG from Telethon (TIGET grant E3), the Italian Ministry of Health (GR-2013-02358956; GR-2016-02364847) and Banca d'Italia (liberal contribution); LN from the Italian Ministry of Health (PE-2016-02363691; E-Rare-3 JTC 2017), the Italian Ministry of University and Research (PRIN 2017 Prot. 20175XHBPN), the EU Horizon 2020 Program (UPGRADE), and the Louis-Jeantet Foundation through the 2019 Jeantet-Collen Prize for Translational Medicine. VV conducted this study as partial fulfillment of her Ph.D. in Molecular Medicine, International Ph.D. School, Vita-Salute San Raffaele University (Milan, Italy). EM conducted this study as partial fulfillment of her Ph.D. in Translational and Molecular Medicine—DIMET, Milano-Bicocca University (Monza, Italy). FF and VL are members of the European Reference Network for Rare Immunodeficiency, Autoinflammatory and Autoimmune Diseases—Project ID No 739543.

## Author contributions

Experiment design, research, data interpretation, and manuscript writing: VV and EM; Human B–T co-culture experiments and data interpretation: GEM; Murine ELISPOT assays, data interpretation, and mouse experiments: MCC; Designing some experiments and data interpretation: GS; Technical support with mouse experiments: LA; Screening of gRNAs and off-target profiling of the selected candidate: CM and FB; Supervision of Screening of gRNAs and off-target profiling of the selected candidate: CC-R; Germinal center analysis: EF; Bioinformatic analysis: SB and IM; Histopathology of *P. murina*-infected mice: AC; Supervision of histopathology of *P. murina*-infected mice: ES; Statistical analyses: PMVR; HIGM1 patients' cells: VL, AP, MK, and AL; Coordination in collection: FF; Experimental design and supervision of HIGM1 mouse experiments and human B–T co-culture experiments: AV; Study design, data interpretation, research supervision, manuscript writing, and work coordination: LN and PG.

## Conflict of interest

LN, PG, GS, and VV are inventors of patent applications on gene editing in HSPC and cell selection owned and managed by the San Raffaele Scientific Institute and the Telethon Foundation, including a patent application on CD40L gene editing shared with Editas Medicine. CM and FB are current employees of Editas Medicine. LN and PG are founders and quota holders of GeneSpire, a startup company aiming to develop gene editing application in HIGM1 and other genetic diseases. All other authors declare no conflict of interests.

## For more information

i   https://genoveselab.dana-farber.org/
ii  https://research.hsr.it/en/institutes/san-raffaele-telethon-institute-for-gene-therapy/gene-transfer-technologies-and-new-gene-therapy-strategies/luigi-naldini.html

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
