## [Review Process File · EMBO Molecular Medicine]

Modelling, Optimization and Comparable Efficacy of T cells and HSC Gene Editing for treating HIGM1

Valentina Vavassori, Elisabetta Mercuri, Genni Marcovecchio, Maria Carmina Castiello, Giulia Schioli, Luisa Albano, Carrie Margulies, Frank Buquicchio, Elena Fontana, Stefano Beretta, Ivan Merelli, Andrea Cappelleri, Paola Rancoita, Vassilios Lougaris, Alessandro Plebani, Maria Kanariou, Arjan Lankester, Francesca Ferrua, Eugenio Scanziani, Cecilia Cotta-Ramusino, Anna Villa, Luigi Naldini, and Pietro Genovese

DOI: 10.15252/emmm.202013545

Corresponding authors: Pietro Genovese (Pietro.Genovese@childrens.harvard.edu) , Luigi Naldini (naldini.luigi@hsr.it)

Review Timeline:

Submission Date:	6th Oct 20
Editorial Decision:	8th Oct 20
Revision Received:	13th Nov 20
Editorial Decision:	1st Dec 20
Revision Received:	8th Dec 20
Accepted:	10th Dec 20

Editor: Zeljko Durdevic

Transaction Report:

(Note: Please note that the manuscript was previously reviewed at another journal and the reports were taken into account in the decision making process at EMBO Molecular Medicine. Since the original reviews are not subject to EMBO's transparent review process policy, the reports and author response cannot be published. With the exception of the correction of typographical or spelling errors that could be a source of ambiguity, letters and reports are not edited. Depending on transfer agreements, referee reports obtained elsewhere may or may not be included in this compilation. Referee reports are anonymous unless the Referee chooses to sign their reports.)

8th Oct 2020

Dear Dr. Genovese,

Thank you for submitting your manuscript to EMBO Molecular Medicine. I have now carefully read your manuscript and discussed it with the other members of our editorial team. In addition, I have also sought external advice on the study from an expert in the field. I am pleased to inform you that we find your manuscript suitable for publication in EMBO Molecular Medicine pending the appropriate revision.

Further consideration of your manuscript will depend on addressing the following points:

- Please revise aims of the study in such way that they are supported by the data presented.
- Please provide more detailed demonstration of gene editing efficiency in stem memory T cells (TSCM). The phenotype definition of TSCM should be demonstrated using more stringent phenotype panel, including CD95, CD11a or other known markers.
- Please perform cellular CD40L and NGFR co-staining to exclude leak expression of CD40L-NGFR construct.
- Please provide a rationale for selecting the construct with the HBB rather than the EF1a splice acceptor.
- Please correct: Reference made to Fig1B in the results section (pg5) does not relate to the data shown in Fig1B.
- Please discuss possible cell doses needed for a therapeutic outcome and clinical utility of the depletion strategy using Cetuximab in regard to the in vitro results.
- Please discuss limitations of the study in regard to a) the missing proof-of-concept that autologous gene edited HSC and/or T cell infusion rescues HIGM1 mouse model, and b) the lack of long-term follow-up study to evaluate the efficiency and safety of the therapy.

Additional experiments that further strengthen the main conclusions of the study are of course appreciated. We would welcome the submission of a revised version within three months for further consideration. However, we realize that the current situation is exceptional on the account of the COVID-19/SARS-CoV-2 pandemic. Please let us know if you require longer to complete the revision.

***** Advice from external expert *****

7 Oct. 2020

I think you should publish this paper. It represents a solid approach to gene editing therapy and appropriate in my opinion for publication in EMBO Molecular Medicine.

Modeling, Optimization and Comparable Efficacy of T cells and Hematopoietic Stem Cells Gene Editing for Treating Hyper IgM Syndrome

We would like to thank the Editor and the Reviewers for their careful and positive evaluation of our manuscript, and the willingness of the Editor to consider a revised version for publication in EMBO Molecular Medicine as an Article. We now provide a revised version of the manuscript according to the Editor requests, including new data and additional clarifications on some of the previously reported findings.

We addressed all the indicated requests in the point-by-point reply below and all changes in the revised manuscript are highlighted in yellow to facilitate the revision.

We hope that the revised version of our study is now suitable for publication and thank the Editor for his consideration.

Further consideration of your manuscript will depend on addressing the following points:

- Please revise aims of the study in such way that they are supported by the data presented.

The aims of the study as well as the limitations imposed by using wild-type cells as surrogate of functional edited cells in the mouse model (see also last point below) were better highlighted in the new introduction and discussion sessions to ensure that all aims are fully supported by the presented data.

- Please, provide more detailed demonstration of gene editing efficiency in stem memory T cells (TSCM). The phenotype definition of TSCM should be demonstrated using more stringent phenotype panel, including CD95, CD11a or other known markers.

We thank the reviewer for highlighting this point. In our original manuscript, we used the minimum panel of surface markers that allows identifying TSCM on in vitro stimulated cells, which operatively may not contain any more naïve cells (Cieri et al., 2013). In order to provide a more stringent demonstration of gene editing efficiency in TSCM, we now expanded our analyses by including the CD95 and CD45RO markers, thus confirm the full TSCM identity as CD95+CCR7+CD45RO+CD62L+ CD45RA+ cells.

CD95 marker distinguishes naïve cells (CD95-) vs TSCM cells (CD95+).

CCR7 marker distinguishes TSCM and CM cells (CCR7+) vs EM and TEMRA cells (CCR7-).

CD45RO marker distinguishes naïve (or circulating, resting TSCM cells) (CD45RO-) vs cultured activated naïve-derived TSCM cells (CD45RO+).

CD62L marker distinguishes TSCM and CM cells (CD62L+) vs EM and TEMRA cells (CD62L-).

CD45RA marker distinguishes TSCM cells (CD45RA+) vs CM cells (CD45RA-).

We used the gating strategy reported in Cieri et al., 2013: in the CD95+ population (almost all the stimulated, live cells were CD95+), we first gated CCR7+ CD45RO+ cells and then CD62L+CD45RA+ cells. Inside the latter double-positive population, we measured NGFR expression as surrogate of gene editing efficiency and compared it with that of total live CD4+ T cells (see new Fig. EV2 B, C). Even by using this more stringent analysis, we observed a mean of 35% gene editing efficiency in the TSCM subpopulation, thus confirming the observation reported in the original manuscript.

Figure EV2 B, C. *B* Representative plots showing more detailed characterization of edited TSCM cells, defined as $CD95^+CCR7^+CD45RO^+CD62L^+CD45RA^+$. *C* Percentage of NGFR+ cells within TSCM cell subpopulation ($CD95^+CCR7^+CD45RO^+CD62L^+CD45RA^+$) or within total live cells, 17 days after CD40LG editing of healthy male donor (HD; $n=3$) derived CD4+ T cells, measured by FACS analysis.

- Please, perform cellular CD40L and NGFR co-staining to exclude leak expression of CD40L-NGFR construct.

As noted in the Reviewer's comment, NGFR expression in edited cells using the selection cassette "showed basal levels of protein expression despite its expression being linked and regulated by the CD40L control elements", which does not allow surface CD40L translocation in absence of T cell stimulation. We hypothesize that this may be due to the physiologically regulated surface exposure of CD40L through trafficking and storage in different secretory compartments than NGFR. To confirm this, and ensuring there is no 'leakiness' in our vector design, we performed intracellular vs surface co-staining of both CD40L and NGFR in absence of stimulation (see new Fig. EV2 F, G). This analysis confirmed that all NGFR+ cells have a CD40L intracellular reservoir in absence of stimulation, which is similar than those measured in unedited and NGFR- cells (please, note that these latter populations contain also cells that are physiologically CD40L negative, thus, as expected, the mean fluorescent intensity in the NGFR+ fraction appears to be slightly higher). As we explain in the manuscript, because upon cell activation CD40L is translocated to the membrane by regulated secretion, its surface expression level might be restored to physiological levels once the stores have been replenished above a certain threshold. These results support the previously reported findings (Casamayor-Palleja et al., 1995; Koguchi et al., 2007) that a second layer of regulation is necessary to mediate CD40L surface translocation, thus further ensuring regulated expression control of CD40L after gene correction with our strategy.

Figure EV2 F, G. *F* Representative plots showing CD40L expression after surface (left) or intracellular staining (right) in UT or bulk edited CD4⁺ T cells derived from male HD in absence of Pma/Ionomycin stimulation. *G* CD40L expression measured by MFI after surface or intracellular staining in UT or bulk edited CD4⁺ T cells derived from male HD in absence of Pma/Ionomycin stimulation (n=3).

- Please provide a rational for selecting the construct with the HBB rather than the EF1a splice acceptor.

To ensure regulated and physiological expression of the CD40LG after editing, we tested different configuration of the donor template on T cells from male donors and used PMA/Ionomycin stimulation to induce CD40L expression. To facilitate this screening, the donor templates were delivered to the cells by integrase defective lentiviral vector (IDLV) or dsODN. Since these delivery vehicles will not need long time-consuming vector production, they allow rapid testing of different template configuration. However, they are not optimal in terms of editing efficiency, especially if compared with AAV6, thus selection of the best performing candidates was performed only based on the expression of the edited CD40LG gene. Since all the splice acceptor tested allowed efficient and complete splice trapping of the endogenous transcript, we selected the HBB because was the first available for AAV6 production. In the revised manuscript, we added more detailed information about the selection process of the template configuration on the legend to Fig EV1.

- Please correct: Reference made to Fig1B in the results section (pg5) does not relate to the data shown in Fig1B.

We thank the Reviewer for spotting this mislabeling. We amended it in the revised manuscript.

- Please discuss possible cell doses needed for a therapeutic outcome and clinical utility of the depletion strategy using Cetuximab in regard to the in vitro results.

We thank the reviewer for highlighting these interesting points, which we both addressed in the revised text and discussion:

1. *The T cell dose used for mouse modeling represents the upper range of cells/kg used in a previous trial with gene edited CD4 T cells (Tebas et al., 2014) and might exceed the doses of pathogen specific adoptive T cell therapies used in HSCT settings (Icheva et al., 2013). While care must be taken when translating results from experimental mouse models to the clinical setting, we should acknowledge that humans, as all wild animals, are exposed to commensal and pathogenic microbes throughout their lives, and this microbiome has a*

profound impact on immune system development, competence and overall health. The use of laboratory mice housed under specific pathogen-free (SPF) conditions is important to improve experimental consistency, but leaves the mice with an underdeveloped immune system (Huggins et al., 2019), thus possibly underestimating the level of immune response to an antigenic challenge predicted for the human setting. Indeed, previous sporadic reports of patients with genetic mosaicism, either an allogeneic HSCT patient with low engraftment (Petrovic et al., 2009) or female carriers with skewed X inactivation in the blood (Hollenbaugh et al., 1994), would support our contention that even low frequencies percentages of CD40L proficient cells, achieved by either T cell or HSPC therapy, are sufficient to provide substantial immune protection. A dose escalation design of a T cell therapy trial will allow safe testing of these predictions in the clinical setting.”

2. *“In vivo depletion of hEGFRt-expressing cells by Cetuximab relies on antibody-dependent cellular cytotoxicity (ADCC), which also requires functional NK cells (Lee et al., 2011). Since ADCC on human cells is difficult to be assessed in xenotransplantation experiments with immunodeficient mice, we explored an in vitro immunotoxin-based strategy to evaluate if edited cells carrying hEGFRt were amenable to antibody-mediated depletion (Palchaudhuri et al., 2016). By culturing edited T cells in the presence of Cetuximab conjugated to the protein synthesis inhibitor toxin saporin (Cetuximab-SAP) or of antibody and toxin alone as controls, we observed substantial depletion (~50%) of hEGFRt-expressing lymphocytes at both doses tested (Fig 3C and D). While the decreased internalization rate of our modified hEGFRt is likely reducing the efficacy of immunotoxin treatment, these data suggest that hEGFRt is a suitable candidate both for in vitro selection and in vivo depletion of CD40LG edited cells”.*

...

“The use of our optimized hEGFRt marker allows coupling selection with the possibility to deplete the transplanted cell product by treatment with a clinically approved monoclonal antibody which, based on the broad clinical experience in tumor therapies, is associated with only minor side effects, such as skin rash (Hansel et al., 2010; Pérez-Soler and Saltz, 2005). While our investigation on human cells remains limited in providing direct evidences of T cell killing, due to the lack of effector cells on xenogeneic models for assessing antibody-dependent cellular cytotoxicity (Shultz et al., 1995; Verma et al., 2017), previous studies performed in full mouse settings have already proved effective depletion from both blood and solid organs of T cell expressing hEGFRt within 4 days after Cetuximab administration (Paszkiwicz et al., 2016; Wang et al., 2011). Indeed, this strategy is already under investigation in several clinical trials as safety control of T cell-mediated cancer immunotherapy (Yu et al., 2019). Nevertheless, since the depletion by Cetuximab remains a relatively slow process, further studies will be necessary to assess whether this approach would also be suitable for controlling more acute adverse events related to T cell administration, such as the cytokine release syndrome reported in some patients after the infusion of activated CD8 T cells.”.

- Please discuss limitations of the study in regard to a) the missing proof-of-concept that autologous gene edited HSC and/or T cell infusion rescues HIGM1 mouse model, and b) the lack of long-term follow-up study to evaluate the efficiency and safety of the therapy.

We thank the Reviewer and Editor for highlighting these points.

- a. The disease rescue experiments in the mouse model were performed using competitive transplants with wild type cells used as surrogate of the corrected cells instead than bona fide gene edited mouse cells. Yet, since the gene editing procedures on mouse HSC or T cells would require significantly different culture protocols and procedures than those

developed and here optimized for clinically ready protocols for human cells, experiments with murine edited cells would provide only limited value in the perspective of future clinical translation. Indeed, the use of specie-specific reagents and the expectedly low efficiency of gene editing achieved on mouse cells would provide limited information on the long-term safety profile of the genetic manipulation on human cells and constrain transplantation studies. Nevertheless, whether edited human cells fully recapitulate upon transplantation the function and long-term persistence of healthy donor cells will have to be determined in clinical studies.

- b. On the contrary, our experiments with mouse cells allow us showing that transferred wild-type T cells can engraft and persist at long-term follow-up in the mouse model (up to 178 days in naïve T cell experiments, up to 219 days in activated T cell experiments, up to 311 days in P. Murina T cell experiment). The impact of gene editing on the fitness of T cells and, consequently, their long-term persistence still need to be clarified. Despite long-term follow-up studies have demonstrated persistence of gene-modified T cells in all memory and effector T cell compartments for up to 14 years (Oliveira et al, 2015) and clinical trials have shown persistence of NHEJ-edited T cells for several months in humans (Tebas et al, 2014; Stadtmauer et al, 2020), no specific studies have been carried out on HDR-edited T cells so far. Nevertheless, if functional immune reconstitution would decrease over time, repeated administrations of the same or a new edited cell product could be performed to prolong therapeutic efficacy.

We have now included these considerations in the discussion our revised manuscript.

1st Dec 2020

Dear Prof. Genovese,

Thank you for the submission of your revised manuscript to EMBO Molecular Medicine. I am pleased to inform you that we will be able to accept your manuscript pending the following final amendments:

1) In the main manuscript file, please do the following:

- Correct/answer the track changes suggested by our data editors by working from the attached/uploaded document.

***** Advice from external expert *****

22 Nov. 2020

I think they have addressed the points adequately.

The authors performed the requested changes.

10th Dec 2020

Dear Prof. Genovese,

We are pleased to inform you that your manuscript is accepted for publication.

YOU MUST COMPLETE ALL CELLS WITH A PINK BACKGROUND ↓
PLEASE NOTE THAT THIS CHECKLIST WILL BE PUBLISHED ALONGSIDE YOUR PAPER

Corresponding Author Name: Pietro Genovese
Journal Submitted to: EMBO Molecular Medicine
Manuscript Number: